# Generation of Site-Specific Accelerograms and Response Spectra Involving Sampling Information from Borehole Records

Yiwei Hu [1,*], Nelson Lam [1], Prashidha Khatiwada [1], Hing-Ho Tsang [2] and Scott Menegon [2]

[1] Department of Infrastructure Engineering, The University of Melbourne, Parkville, VIC 3010, Australia
[2] Department of Civil and Construction Engineering, Swinburne University of Technology, Hawthorn, VIC 3122, Australia
*   Correspondence: huyh1@student.unimelb.edu.au

**Abstract:** This paper is aimed at serving the needs of structural engineering designers of an important structure (or a group of structures located on the same site) who is seeking guidance on how to obtain accelerograms and/or derive response spectra that accurately represent the site subsoil conditions as informed by the borelogs. The presented site-specific seismic action model may be used to replace the default seismic action model stipulated for the designated site class. Presented in this article is a procedure for generating soil surface motions in an earthquake, and their associated site-specific response spectra, taking into account details of the soil layers. Dynamic site response analyses are involved. The conditional mean spectrum methodology is employed for selecting and scaling accelerograms for obtaining input motion on bedrock. The selection depends on the natural period of both the site and the structure. Multiple borelogs taken from within the same site are analysed to identify the critical soil column models without having to conduct site response analysis on every borelog. The technique for simplifying the soil layers utilising the shear strain profile is introduced to further cut down on the time of analyses. The procedures described in this article have been written into a web-based program that is freely accessible to engineering practitioners.

**Keywords:** site-specific response spectra; intraplate regions; structural design; site response analysis; simplified soil column model

## 1. Introduction

The earthquake actions are typically estimated with code-stipulated elastic response spectrum models for different soil classes. The code response spectrum models are easy to implement, but have limitations for two main reasons. First, a code response spectrum model is derived by enveloping response spectra associated with a diversity of earthquake scenarios, some of which may not be applicable in specific instances. Second, the statistical analyses of data for deriving a code response spectrum model can under-represent the actual extent of site amplification. A more realistic representation of earthquake actions is site-specific response spectra or a suite of soil surface ground motions developed explicitly for the construction site. However, this procedure involves regional seismic hazard analyses, soil condition analyses, and site response analyses, which require extensive input information and insights into earthquake characteristics and soil behaviours. Guidelines or facilities for generating site-specific response spectra in accordance with the design code are not available to engineering practitioners in Australia.

This paper is aimed at presenting detailed descriptions of a procedure for generating ground motions and response spectra on the soil surface of a targeted site for a range of projected earthquake scenarios for use in structural design and assessment. The soil modification behaviour, which is the increase in earthquake wave amplitudes when travelling from bedrock to the softer soils, is obtained from site response analysis of representative

soil column models of the targeted site as derived from borelogs taken from subsurface site investigations. This approach for modelling soil modification behaviour has the merit of taking into account details of the soil layers and their dynamic properties. Detailed guidance in relation to the conversion of standard penetration blow counts into shear wave velocity values of each soil layer is presented.

The procedure should provide a more accurate evaluation of the potential site hazard than the widely adopted code approach of determining the design response spectrum of a site based on broad site classifications [1–3]. The input motions transmitted from the bedrock should have the frequency contents that represent real earthquake events, and are to be derived by adopting the conditional mean spectrum (CMS) methodology as introduced in a companion article of the special issue [4]. A scheme of selecting input motions in accordance with the natural period of both the site, and that of the structure, is presented.

Site characterisation of an accelerogram recording station is typically without the support of information sourced from multiple boreholes taken from the same site. Neglecting intra-site variability is believed to have contributed to discrepancies between the recorded and simulated soil surface motions. The presence of intra-site variability means that a soil column model representing the subsurface conditions of a soil site must not be based on only a single borelog. In the proposed procedure, soil layer details taken from multiple boreholes drilled on the same site need to be analysed and sorted to construct soil column models that give conservative predictions of the soil surface response in an earthquake event. A technique of simplifying a soil column model to economise on computational time is also presented.

The methodology for generating site-specific response spectra and accelerograms in engineering practice comprises three routines as stipulated in the American code ASCE16 [2]:

(1) Interpretation and analysis of information presented in a borelog for estimating the shear wave velocity (*SWV*) profile and dynamic properties of the soil layers;
(2) Selection and scaling of accelerograms for defining the input motions transmitted from the bedrock;
(3) Identification of the critical soil column models and execution of site response analysis for generating accelerograms and response spectra on the soil surface.

The three routines are discussed in Sections 2–4. The case study based on Australian conditions is then presented as an example for illustration (Section 5).

## 2. Interpretation and Analysis of Information Presented in a Borelog

This section presents detailed descriptions of the procedures for constructing soil column models based on analysis of site-specific information as presented in the borelog. The site investigation record as presented in a borelog contains descriptions of the soil characteristics (soil type and moisture content) and quantitative test data. Routine 1 is for inferring information reported in the borelog to derive the following: (1) the *SWV* profile, (2) the density profile, (3) material curves characterising soil behaviour in seismic conditions, and (4) the initial site natural period. Table 1 summarises the input and output parameters associated with this routine.

**Table 1.** Input and output for Routine 1: interpretation and analysis of information presented in a borelog.

| Input | Output |
|---|---|
| <ul><li>Thickness of each layer ($H_i$) [1]</li><li>Standard penetration test (SPT) blow count of each layer ($N_{measured, i}$) [1]</li><li>Energy ratio</li><li>Bedrock shear wave velocity</li><li>Soil type (clay/silt/sand/gravel)</li><li>Soil age, moisture content, and plasticity index of each layer (optional)</li></ul> | <ul><li>*SWV* profile</li><li>Density profile</li><li>Initial site period</li><li>Material curves for each layer</li></ul> |

[1] subscript *i* denotes layer number.

### 2.1. Modelling Shear Wave Velocity and Density Profile

One of the main causes of the soil amplification phenomenon is the impedance contrast between the bedrock and the overlying soil medium. The impedance of the medium is calculated as the product of *SWV* and density. Both media need to be modelled from information inferred from the borelog. Many empirical models have been developed to correlate the *SWV* to the SPT blow counts. This study employs the Imai and Tonouchi model [5], which contains correlating relationships for a series of soil types and ages of the geological formation (abbreviated herein as "soil ages") as listed in Table 2. The parameter $N_{60}$ in the listed equations is the normalised SPT blow count corrected to a constant energy transfer ratio of 60%.

$$N_{60} = Energy\ Ratio \times N_{measured} \tag{1}$$

where *Energy Ratio* is the actual hammer energy transfer ratio divided by the standard ratio of 60%; $N_{measured}$ is the raw SPT blow counts as recorded from in situ testing.

**Table 2.** *SWV* and SPT blow count correlation equations in the Imai and Tonouchi model [5].

| Soil Age [1] | Soil Type | Correlation Equation | Equation No. |
|---|---|---|---|
| Holocene | Clay and silt | $SWV = 103.8 \times N_{60}^{0.27}$ | (2) |
| | Sand | $SWV = 85 \times N_{60}^{0.29}$ | (3) |
| | Gravel | $SWV = 72.3 \times N_{60}^{0.35}$ | (4) |
| Pleistocene | Clay and silt | $SWV = 124.4 \times N_{60}^{0.26}$ | (5) |
| | Sand | $SWV = 106.6 \times N_{60}^{0.29}$ | (6) |
| | Gravel | $SWV = 132.4 \times N_{60}^{0.25}$ | (7) |

[1] In situation where soil age is not specified in the borelog, *SWV* is calculated as the average value from the equations for holocene and pleistocene soils.

In addition to the Imai and Tonouchi model, two more models, namely the Ohta and Goto model [6] and the PEER model [7], have been incorporated into the Quake Advice website for free online access. Refer to Hu, et al. [8], should further details on the SPT($N_{60}$)-*SWV* conversion be required.

Table 3 provides the estimated density values for different soil types and ranges of SPT blow counts. More accurate estimations are presented in Appendix A, which provides information for the determination of soil division and moisture content.

**Table 3.** Soil density estimated from soil type and SPT blow counts.

| Sand | | Gravel | | Silt/Clay | |
|---|---|---|---|---|---|
| $N_{60}$ | Density (kg/m³) | $N_{60}$ | Density (kg/m³) | Soil Type | Density (kg/m³) |
| 0–4 | 1760 | 0–4 | 1950 | Low plasticity silt | 1570 |
| 4–10 | 1810 | 4–10 | 1990 | High plasticity silt | 1660 |
| 10–30 | 1900 | 10–30 | 2050 | Low plasticity clay | 1500 |
| 30–50 | 2010 | 30–50 | 2120 | Medium plasticity clay | 1560 |
| >50 | 2070 | >50 | 2160 | High plasticity clay | 1640 |

The conversion from a borehole record to a *SWV* and density profile is illustrated by an example as presented in Figure 1. The bedrock *SWV* ($V_R$) of the example site was taken to be 800 m/s. The bedrock density ($\rho_R$) was computed accordingly using Equation (8) as recommended by Tsang and Pitilakis [9]. Refer to Appendix B for the detailed borelog data and results of the conversion.

$$\rho_R = \left(1.8 + \frac{V_R}{3550}\right) \times 1000 \tag{8}$$

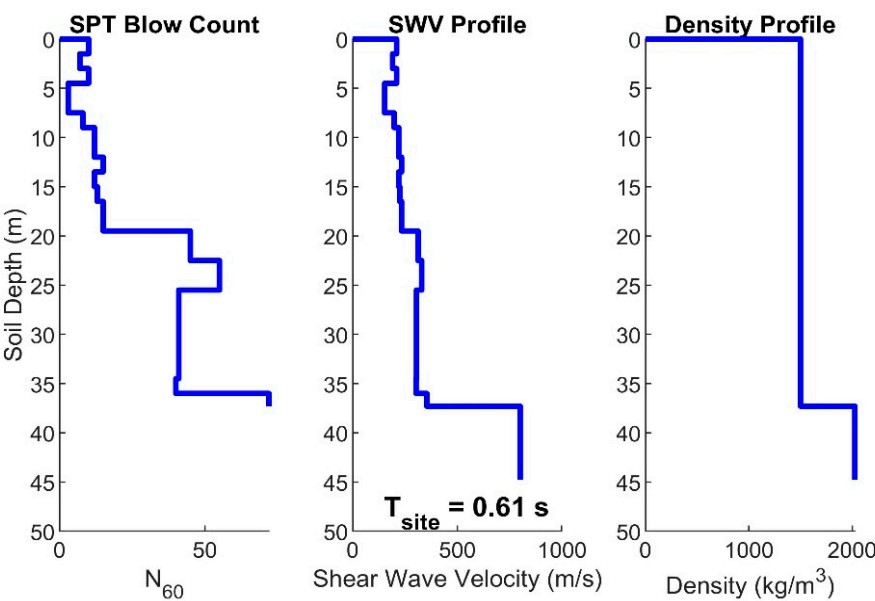

**Figure 1.** Shear wave velocity and density profile estimated from borehole data.

The initial site period ($T_{site}$) can be estimated as four times the time taken for the seismic waves to be transmitted from the bedrock to the soil surface as per Equation (9) [1].

$$T_{site} = \sum_{i=1}^{N} \frac{4H_i}{SWV_i} \tag{9}$$

where *i* denotes the soil layer number; $H_i$ is the thickness of layer *i*; $SWV_i$ is the shear wave velocity of layer *i*; and *N* is the total number of soil layers in the soil column.

### 2.2. Modelling Dynamic Property

Soil sediment exhibits nonlinear behavior when subject to earthquake loadings. With increasing shear deformation (represented by the shear strain in the soil), the stiffness of the soil layer is reduced and the amount of energy dissipation is increased. The correlating relationships between shear strain ($\gamma$), secant shear modulus ($G_{sec}$), and damping ratio ($\zeta$) are empirically determined from material curve models. The Hardin and Drnevich model [10] presents material curves based solely on the soil plasticity index (*PI*) as represented by Equations (10)–(13). The *PI*-dependent reference shear strain ($\gamma_{ref}$) values are summarised in Table 4.

$$\frac{G_{sec}}{G_{max}} = \frac{1}{1 + \frac{\gamma}{\gamma_{ref}}} \tag{10}$$

$$\zeta = \zeta_{initial} + \zeta_{max} \frac{\left(\frac{\gamma}{\gamma_{ref}}\right)}{\left(1 + \frac{\gamma}{\gamma_{ref}}\right)} \tag{11}$$

$$\zeta_{max} = 0.16 - 0.001 \times PI(\%) \geq 0 \tag{12}$$

$$\zeta_{initial} = 0.015 + 0.0003 \times PI(\%) \leq 0.058 \tag{13}$$

where $\gamma$ is shear strain; $\gamma_{ref}$ is the *PI*-dependent reference shear strain; $\zeta$ is damping ratio; $\zeta_{initial}$ and $\zeta_{max}$ are the *PI*-dependent initial and maximum damping ratio, respectively; $G_{sec}$ and $G_{max}$ are the secant and maximum shear modulus, respectively; $\frac{G_{sec}}{G_{max}}$ is the shear modulus reduction ratio.

**Table 4.** The reference shear strain values for the Hardin and Drnevich model.

| *PI* (%) | 0 | 15 | 30 | 45 |
|---|---|---|---|---|
| $\gamma_{ref}$ [1] (%) | 0.0025 | 0.0045 | 0.1 | 0.2 |

[1] Use linear interpolation where necessary to obtain the reference shear strain for any value of PI.

The material dynamic properties are controlled by the value of *PI*, as illustrated by the three curves corresponding to the *PI* value of 0%, 15%, and 30% (Figure 2). The *PI* values can either be inferred directly from information presented in the borelog or be estimated from descriptions of the soil (in the case of cohesive soils) in accordance with the local code of practice. For example, the *PI* value of low plasticity clay may be taken as 10% based on the Australian code for geotechnical site investigation: AS1726:2017 [11]. In the absence of such descriptions, *PI* = 0% may be assumed for sand and gravel, whereas *PI* = 30% may be taken for silt and clay.

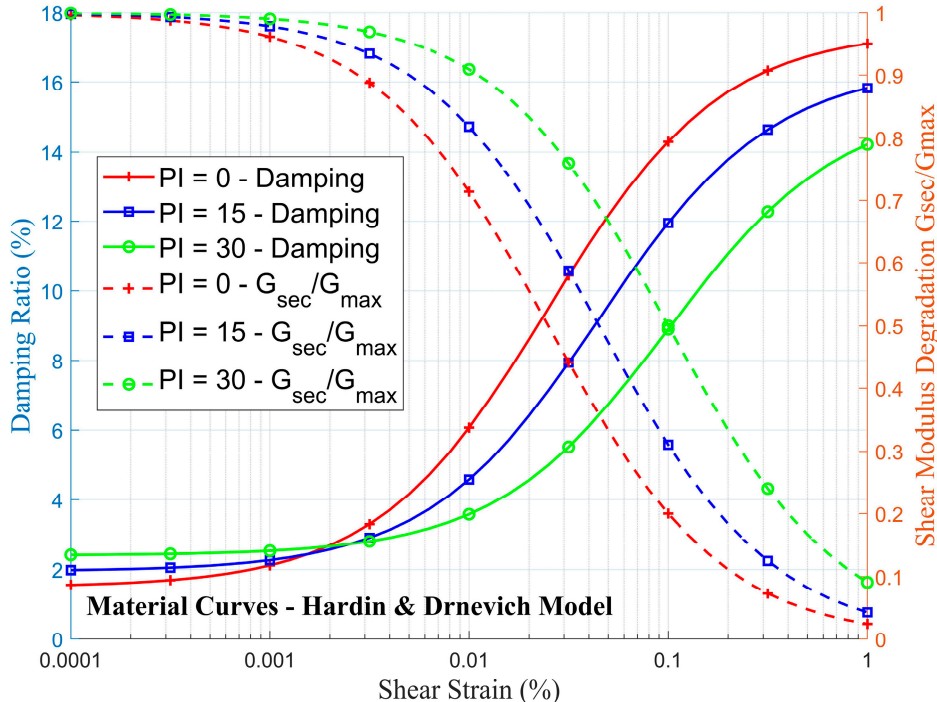

**Figure 2.** Estimation of damping ratio curves and shear modulus degradation curves based on the Hardin and Drnevich Model for plasticity index of 0%, 15%, and 30%.

In addition to the Hardin and Drnevich model, the Quake Advice website has implemented three more material curve models, namely the Vucetic and Dobry model [12], the Darendeli model [13], and the Zhang et al. model [14]. The latter two require the estimated vertical pressure on top of the soil surface induced by the structure.

### 3. Selection and Scaling of Accelerograms for Defining Input Bedrock Motion

The motions at the bedrock level for input into site response analyses can be selected and scaled to the CMS as recommended in publications by Baker and co-workers [15–17]. The application of the CMS methodology in intraplate regions of lower seismicity has been studied by Hu and co-workers to overcome challenges associated with the lack of representative authentic strong motion data that is available [18]. Part of the contents presented in this section overlaps with the companion paper [4] in order to make the current paper self-contained. The selection of input motions for use in site response analyses based on the natural period of both the site and the structure to ensure good coverage of the contributing scenarios as

presented in the later part of this section is original. Table 5 summarises the input and output parameters for implementing the CMS methodology.

**Table 5.** Input and output for Routine 2: on selection and scaling of accelerograms for defining input motion transmitted onto the bedrock.

| Input | Output |
|---|---|
| • Code response spectrum on rock sites for a given return period (e.g., 2500 years) <br> • Initial site period (from Routine 1) <br> • Fundamental period of vibration of the structure. | • A set of CMS based on matching the code response spectrum at four distinctive reference periods (i.e., 0.2, 0.5, 1, and 2 s) <br> • An ensemble (24) of accelerograms on bedrock <br> • An ensemble (12–16) of accelerograms on bedrock that are highlighted for nonlinear time history analysis. |

With the CMS methodology, the target response spectrum for sourcing ground motions is based on a specific earthquake event that is critical to the structure at a pre-defined reference period ($T^*$). The CMS is, therefore, a more realistic representation of earthquake characteristics than the code stipulated design spectrum, which is essentially based on enveloping a range of earthquake scenarios as opposed to a specific projected scenario. Routine 2 is to construct four CMS to define the bedrock motions for four reference periods: 0.2, 0.5, 1, and 2 s. Six 'best match' ground motion records are to be selected and scaled for each reference period, totalling twenty-four input motions transmitted from the bedrock. For each reference period, the construction of the CMS follows a six-step procedure as outlined in the following:

(1) Construct the code response spectrum on rock sites $Sa(T)$ for a given return period, and identify the spectral amplitude $Sa(T^*)$ at the reference period;

(2) Determine a set of representative ground motion prediction expressions (GMPEs) and weighting factors for determining ground motion parameters for given magnitude–distance (M–R) combinations (along with other parameters). The set of GMPEs may comprise locally derived empirical GMPEs and generic GMPEs derived from stochastic simulations of the seismological model [19–21];

(3) Identify the main contributing earthquake scenario expressed as an M–R combination by de-aggregation analysis [22–24];

(4) Estimate from the GMPEs the weighted median response spectral acceleration $\mu(T)$ and standard deviations $\sigma(T)$ for the contributing scenario;

(5) Calculate the value of epsilon $\varepsilon(T^*)$ for scaling up the median response spectral amplitude to match the code spectral amplitude at a given reference period $T^*$;

$$\varepsilon(T^*) = \frac{\log(Sa(T^*)) - \log(\mu(T^*))}{\sigma(T^*)} \quad (14)$$

(6) Construct the CMS by applying a period-dependent correlation coefficient $\rho(T, T^*)$. The correlation coefficient equals 1 at $T = T^*$ and values less than 1 at other periods.

$$Sa_{CMS}(T) = \exp(\log(\mu(T)) + \sigma(T) \times \varepsilon(T^*) \times \rho(T, T^*)) \quad (15)$$

$$\rho(T, T^*) = 1 - \cos\left(\frac{\pi}{2} - \left(0.359 + 0.163 I_{(T_{min} < 0.189)} \ln\frac{T_{min}}{0.189}\right) \ln\frac{T_{max}}{T_{min}}\right) \quad (16)$$

where $Sa_{CMS}$ is the spectral value of the CMS; $T_{min}$ is the lower of $T^*$ and $T$; $T_{max}$ is the higher of $T^*$ and $T$; $I_{(T_{min} < 0.189)}$ is the indicator that is equal to unity if $T_{min} < 0.189$ and is equal to zero otherwise.

Figure 3 presents an example CMS at reference period of $T^* = 0.5$ s as shown by the red line, generated in compliance with the Australian code response spectrum for

rock sites with a 2500-year return period. Five GMPEs [21,25–28] were employed for the construction of the CMS, each carrying a 20% weight. The controlling earthquake scenario was a magnitude 6 earthquake at a site-source distance of 23 km. The CMS was found to match the code spectrum at $T^* = 0.5\ s$ and take lower values at periods other than the reference period.

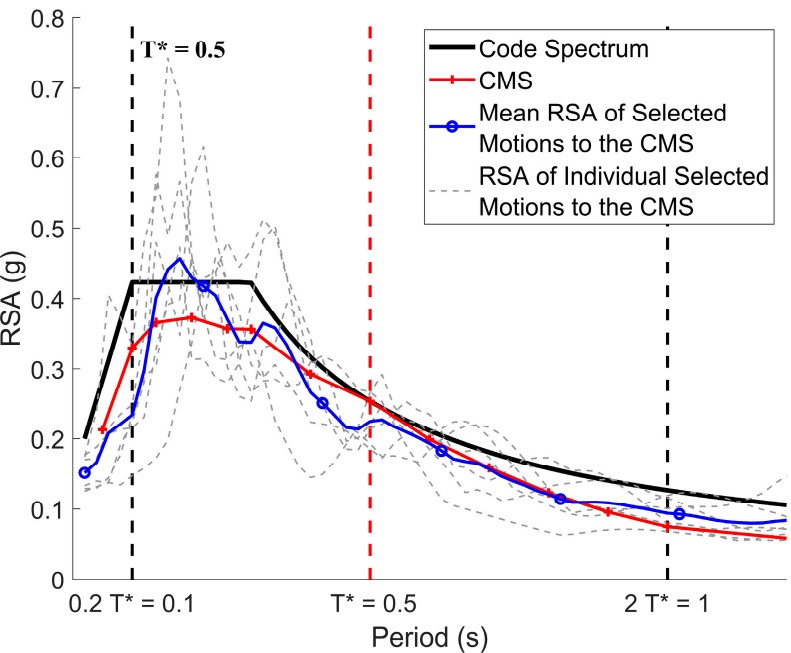

**Figure 3.** The CMS at $T^* = 0.5$ s in compliance with the Australian code response spectrum for rock sites and the acceleration response spectra of the sourced motions.

Following the determination of the target spectra, ground motion accelerograms were sourced from the international ground motion database of NGA-West2 hosted by the Pacific Earthquake Engineer Research Centre [29]. The magnitude and source-site distance of the earthquake events were specified as the main contributing earthquake scenario for the pre-defined return period (refer to step 3 of the procedure for constructing the CMS). The search criteria for retrieving ground motions from the NGA-West2 database are listed in below.

- Style of faulting: reverse/oblique (typical of intraplate earthquakes);
- Magnitude: magnitude range (half-bin width) of ±0.3 $M_w$ centered at the magnitude of the controlling scenarios;
- Joyner-Boore distance Rjb (distance to the fault projection to the surface): distance range (half-bin width) of ± 30 km centered at the distance of the controlling scenarios (with the range extended to ± 50 km at $T^* = 2$ s);
- $V_{S,30}$: 450 ms$^{-1}$ to 1800 ms$^{-1}$ representing rock conditions.

Ground motions that met the search criteria were scaled to match with the spectral values of the CMS over the natural period range: 0.2 $T^*$ to 2 $T^*$. The scaling factor was calculated based on the stronger of the two horizontal components of the record [18] using Equation (17) [15].

$$Scaling\ Factor = \frac{\sum_{i=1}^{n} Sa_{CMS}(T_i)}{\sum_{i=1}^{n} Sa_0(T_i)} \tag{17}$$

where $Sa_0$ is the amplitude of the spectrum for the individual motion before scaling; $T_1$ and $T_n$ are equal to 0.2 $T^*$ to 2 $T^*$, respectively.

The misfit between the target CMS and the scaled motions is represented by the mean squared error (MSE) as calculated using Equation (18). Six scaled motions with the smallest misfits were selected for each of the four reference periods, totalling twenty-four

code-compliant bedrock motions. On the condition that the ensemble contained multiple earthquake records of the same event but documented at different stations, the repeated records were to be replaced by alternatives that were close to the target spectrum. Using records sourced from different events has the benefit of diversity in the ensemble.

$$MSE = \frac{1}{n} \sum_{i=1}^{n} (\ln Sa_{scaled}(T_i) - \ln Sa_{CMS}(T_i))^2 \qquad (18)$$

where $Sa_{scaled}$ is the amplitude of the spectrum for individual motion after scaling; $T_1$ and $T_n$ are equal to 0.2 $T^*$ and 2 $T^*$, respectively.

The ensemble of 24 ground motions at the bedrock level was input into site response analysis for simulating soil surface motions and site-specific response spectra. Depending on the fundamental period of the structure ($T_{structure}$) and the initial site natural period ($T_{site}$), 12–16 bedrock motions that were more likely to control the design of the structure were reserved for use in nonlinear time history analysis.

The selection scheme as proposed by Hu and co-workers [8] recommends selecting additional ground motions at reference periods that are close to the site period or the building period; a minimum of two records should be selected at each reference period to cover a broad period range in order to capture the influence of period shifts and higher modes developed in the structure. The selection scheme stipulates that should either $T_{structure}$ or $T_{site}$ be within ±20% of one of the four reference periods, six records shall be selected for that reference period; should either $T_{structure}$ or $T_{site}$ be between two reference periods, but not within ±20%, then at least four records shall be selected for the two adjacent reference periods. The selection scheme as described above is presented diagrammatically in Figure 4.

To present the outcome of the ground motion selection employing the procedure as described, the authors sourced an ensemble of earthquake records in compliance with the Australian code response spectrum for rock sites with a 2500-year return period. The earthquake information is summarised in Table 6; the acceleration response spectra of the six bedrock motions for reference period $T^* = 0.5$ s are presented in Figure 3. Assuming the building period $T_{structure}$ is 1 s and the site period $T_{site}$ is 0.61 s, the highlighted records for nonlinear time history analysis are accelerogram nos. 1–2, 7–10, and 13–20 following the selection scheme. This case fits in scenario (f) as presented in Figure 4, where $T_{structure}$ is within 20% of the reference period $T^* = 1$ s and $T_{site}$ is between $T^* = 0.5$ and 1 s. Hence, two accelerograms shall be selected from the $T^* = 0.2$ s group; six accelerograms from the $T^* = 0.5$ s group; four accelerograms from the $T^* = 1$ s group; and two accelerograms from the $T^* = 2$ s group.

**Table 6.** Listing of the twenty-four accelerograms for Australian conditions with a return period of 2500 years.

| Accelerogram Ref. Number | Earthquake Name | Reference Periods (s) | Year | Station Name | Magnitude | Rjb (km) | Peak Ground Acceleration (g) | Scaling Factor |
|---|---|---|---|---|---|---|---|---|
| 1 | Whittier Narrows-02 | 0.2 | 1987 | Mt Wilson—CIT Seis Sta | 5.27 | 16.4 | 0.175 | 1.21 |
| 2 | Northridge-06 | 0.2 | 1994 | Beverly Hills—12,520 Mulhol | 5.28 | 10.6 | 0.130 | 0.85 |
| 3 | Christchurch—2011 | 0.2 | 2011 | PARS | 5.79 | 8.5 | 0.126 | 0.61 |
| 4 | Sierra Madre | 0.2 | 1991 | Cogswell Dam—Right Abutment | 5.61 | 17.8 | 0.151 | 0.50 |
| 5 | Friuli (aftershock 9)_Italy | 0.2 | 1976 | San Rocco | 5.5 | 11.9 | 0.127 | 1.41 |
| 6 | Lytle Creek | 0.2 | 1970 | Wrightwood—6074 Park Dr | 5.33 | 10.7 | 0.215 | 1.06 |
| 7 | Christchurch—2011 | 0.5 | 2011 | GODS | 5.79 | 9.1 | 0.175 | 0.63 |

**Table 6.** *Cont.*

| Accelerogram Ref. Number | Earthquake Name | Reference Periods (s) | Year | Station Name | Magnitude | Rjb (km) | Peak Ground Acceleration (g) | Scaling Factor |
|---|---|---|---|---|---|---|---|---|
| 8 | Chi-Chi_ Taiwan-05 | 0.5 | 1999 | HWA031 | 6.2 | 39.3 | 0.128 | 1.91 |
| 9 | Chi-Chi_ Taiwan-05 | 0.5 | 1999 | HWA005 | 6.2 | 32.7 | 0.124 | 1.46 |
| 10 | Whittier Narrows-01 | 0.5 | 1987 | Pacoima Kagel Canyon | 5.99 | 31.6 | 0.169 | 1.04 |
| 11 | Chi-Chi_ Taiwan-03 | 0.5 | 1999 | CHY041 | 6.2 | 40.8 | 0.132 | 1.00 |
| 12 | N. Palm Springs | 0.5 | 1986 | Anza—Red Mountain | 6.06 | 38.2 | 0.171 | 1.77 |
| 13 | Chi-Chi_ Taiwan-06 | 1 | 1999 | CHY041 | 6.3 | 45.7 | 0.094 | 0.53 |
| 14 | Northridge-01 | 1 | 1994 | LA—Temple and Hope | 6.69 | 28.8 | 0.113 | 0.62 |
| 15 | Coalinga-01 | 1 | 1983 | Parkfield—Fault Zone 11 | 6.36 | 27.1 | 0.084 | 1.08 |
| 16 | Coalinga-01 | 1 | 1983 | Parkfield—Stone Corral 3E | 6.36 | 32.8 | 0.170 | 1.13 |
| 17 | San Fernando | 1 | 1971 | Lake Hughes #4 | 6.61 | 19.4 | 0.198 | 1.27 |
| 18 | Chi-Chi_ Taiwan-06 | 1 | 1999 | WHA019 | 6.3 | 52.4 | 0.087 | 1.68 |
| 19 | Loma Prieta | 2 | 1989 | SF—Diamond Heights | 6.93 | 71.2 | 0.076 | 0.67 |
| 20 | Chuetsu-Oki_ Japan | 2 | 2007 | NGN004 | 6.8 | 78.2 | 0.072 | 1.80 |
| 21 | Chuetsu-Oki_ Japan | 2 | 2007 | NGNH28 | 6.8 | 76.7 | 0.051 | 1.42 |
| 22 | Iwate_ Japan | 2 | 2008 | AKT009 | 6.9 | 119 | 0.086 | 1.66 |
| 23 | Loma Prieta | 2 | 1989 | Berkeley—Strawberry Canyon | 6.93 | 78.3 | 0.077 | 1.01 |
| 24 | Chuetsu-Oki_ Japan | 2 | 2007 | NGNH27 | 6.8 | 91.4 | 0.050 | 1.29 |

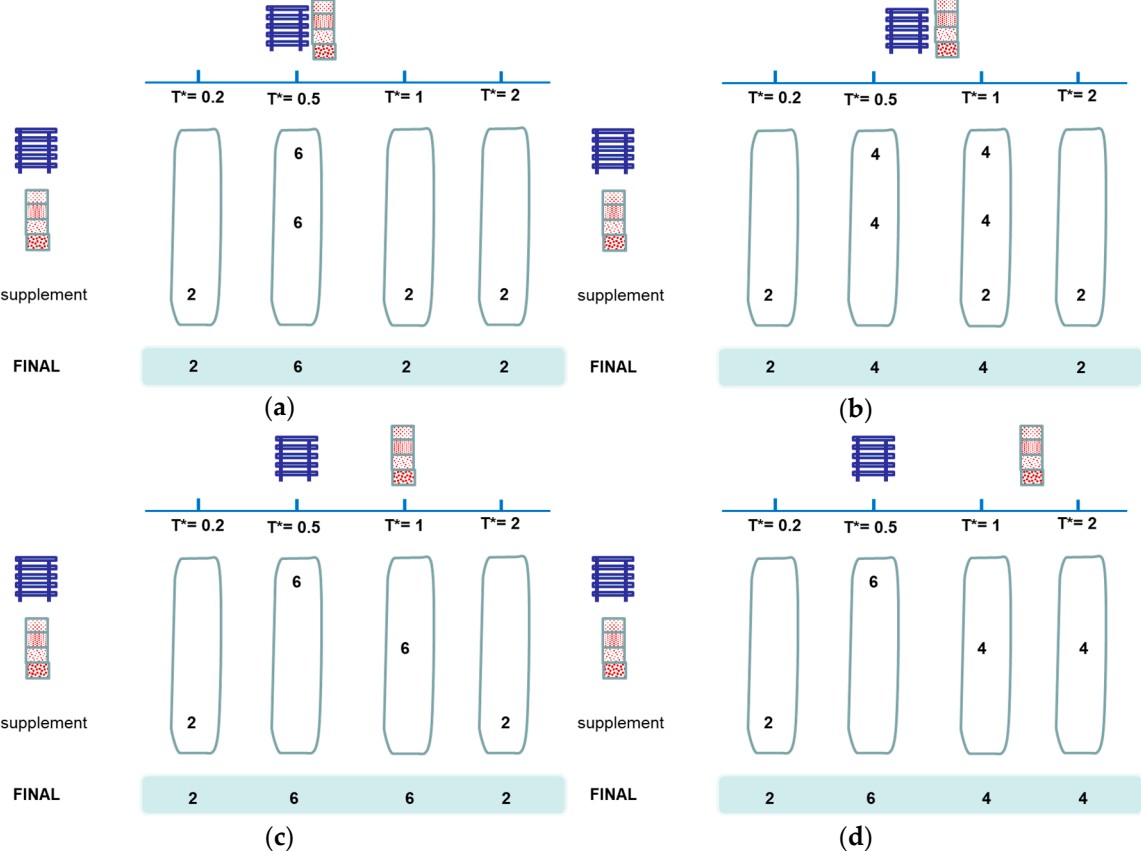

**Figure 4.** *Cont.*

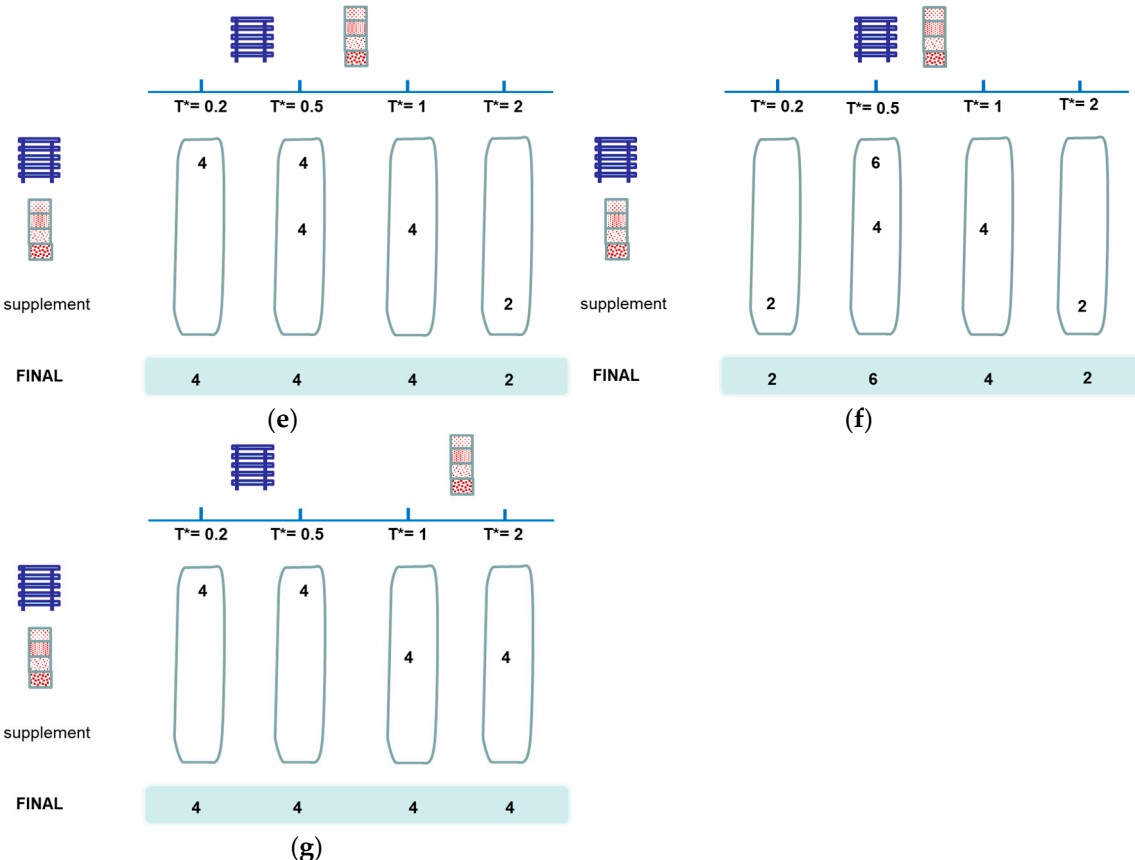

**Figure 4.** The selection scheme to identify 12–16 ground motions from the ensemble of 24 records: (**a**) $T_{site}$ and $T_{structure}$ are close and within 20% of one reference period; (**b**) $T_{site}$ and $T_{structure}$ are close and not within 20% of any reference period; (**c**) $T_{site}$ and $T_{structure}$ are separate and are within 20% of two different reference periods; (**d**) $T_{site}$ and $T_{structure}$ are separate; one is within 20% of a reference period and another is between two other reference periods; (**e**) $T_{site}$ and $T_{structure}$ are not within 20% of any reference periods and are in adjacent intervals; (**f**) $T_{site}$ and $T_{structure}$ are close; one is within 20% of a reference period and another is between that and another reference period; (**g**) one of $T_{site}$ and $T_{structure}$ is between 0.24 and 0.4 s, and another is between 1.2 and 1.6 s.

## 4. Execution of Site Response Analysis of the Critical Soil Column Models

With the soil column models and input bedrock motions determined using Routine 1 and 2, site-specific response spectra and soil surface motion accelerograms can be generated by the use of equivalent linear analysis of the soil column models [30]. The computational cost depends on the number of input motions, the number of soil column models, and the number of soil layers in each soil column model. Considering processing 24 input motions at the bedrock level, the amount of time taken to simulate soil surface motions can be enormous when multiple borehole records are taken from the same site (which is common in engineering practice). Routine 3, which incorporates a borehole information sampling scheme, aims at operating site response analysis with the most conservative soil column models only. A summary of the input and output parameters associated with Routine 3 is presented in Table 7.

**Table 7.** Input and output for Routine 3: execution of site response analysis of the critical soil column models for generating accelerograms and response spectra on the soil surface.

| Input | Output |
|---|---|
| • Bedrock accelerogram ensemble (from Routine 2) <br> • Soil column models (from Routine 1) <br> • Fundamental period of vibration of the structure. | • Identification of the critical (most conservative) soil column models <br> • An ensemble (24) of site-specific response spectra on soil surface <br> • An ensemble (12–16) of soil surface accelerograms. |

*4.1. Sampling the Critical Soil Columns Taken from the Same Site*

Borelogs taken from the same site may have differing details of the soil layers, resulting in inconsistent predictions of the soil surface motions and response spectra. When the soil-surface-to-bedrock depths of all borelogs are nearly identical, the effect of site horizontal heterogeneity is minor. The borehole record that results in the highest amplification ratio at the fundamental natural period of the structure is critical for the structural assessment. The conventional approach to the computation would involve site response analysis systematically covering all the soil column models and input motions. The proposed more efficient and cost-saving procedure is to first identify the critical soil columns models based on the dynamic soil properties and the predicted intensity of earthquake ground shaking. The first step of the procedure is to estimate the shear strain profile that is compatible with the bedrock motion ensemble for each soil column model. An outline of the iterative procedures for determining the shear strain profile involves the following steps:

(1) Calculate the average velocity response spectra of the 12–16 bedrock motions as sourced in Routine 2;

(2) Assign initial values for the proposed effective shear strain of each soil layer ($\gamma_{eff,i}$) by $10^{-5}\%$ to represent the low level of shear strains, with subscript *i* denoting the soil layer number;

(3) Determine the strain-compatible damping ratio ($\zeta_j$) and shear modulus reduction ratio ($G_{red,i} = G_{sec,i} / G_{max,i}$) for each soil layer based on the material curves as determined using Routine 1;

(4) Calculate the reduced *SWV* values, the shifted modal periods, and ratio of the impedance contrast by Equations (19)–(22).

$$V_i = SWV_i \times \sqrt{G_{red,i}} \tag{19}$$

$$T_{site,1} = \sum_{i=1}^{N} \frac{4H_i}{V_i} \tag{20}$$

$$T_{site,2} = \frac{T_{site,1}}{3} \tag{21}$$

$$\alpha = \frac{\rho_s V_s}{\rho_r V_r} \tag{22}$$

where *N* is the total number of soil layers; layer *N* refers to the soil layer immediately above the bedrock; $T_{site,1}$ and $T_{site,2}$ are the shifted first and second modal period of the soil column, respectively; $\alpha$ is the impedance contrast between the bedrock and the soil medium; $\rho_r$ and $V_r$ are the bedrock density and shear wave velocity, respectively; $\rho_s$ and $V_s$ are the averaged soil density and shear wave velocity as defined by Equations (23) and (24), respectively.

$$\rho_s = \frac{\sum_{i=1}^{N} \rho_i \times H_i}{\sum_{i=1}^{N} H_i} \tag{23}$$

$$V_s = \frac{\sum_{i=1}^{N} H_i}{\sum_{i=1}^{N} H_i / V_j} \tag{24}$$

(1) Determine the maximum shear strain developed in the bedrock for the first two modes of vibration of the soil column.

$$\gamma_{max,bedrock,j} = (-1)^j \times \frac{4}{\pi(2j-1)} \times \frac{RSV_{T_{site,j}}}{V_r} \times \sqrt{\frac{7}{\zeta + 2}} \times \alpha \tag{25}$$

where $j$ denotes the modal number, $j = 1\&2$ refers to the numbering of the first two modes; $\gamma_{bedrock,max,j}$ is the estimated maximum bedrock shear strain for the $j$th mode; $RSV_{T_{site,j}}$ is the averaged bedrock response spectral velocity (as determined from step 1) for the $j$th modal period; $\zeta$ is the averaged damping ratio in percentages as defined by Equation (26).

$$\zeta = \frac{\sum_{i=1}^{N} (\zeta_i \times H_i \times \gamma_{eff,i}^2 \times \rho_i \times V_i^2)}{\sum_{i=1}^{N} \left(H_j \times \gamma_{eff,i}^2 \times \rho_i \times V_i^2\right)} \tag{26}$$

(2) Calculate the maximum shear strain for each soil layer. The calculation begins with the soil layer, which is immediately above the bedrock (layer $N$) and continues to the other soil layers from bottom up.

$$\gamma_{max,i,j} = \frac{\rho_{i+1} V_{i+1}^2}{\rho_i V_i^2} \times \frac{\sin(\theta_{i,j}) - \frac{\theta_{i,j}}{\cos(\theta_{i,j})} \zeta_i^2}{\sin(\theta_{i+1,j}) - \frac{\theta_{i+1,j}}{\cos(\theta_{i+1,j})} \zeta_{i+1}^2} \times \gamma_{max,i+1,j} \tag{27}$$

$$\theta_{i,j} = \frac{\pi}{2} \frac{T_i}{T_{site,1}} (2j-1) \tag{28}$$

$$T_i = \sum_{m=1}^{i} \frac{4H_m}{V_m} - \frac{2H_i}{V_i} \tag{29}$$

where $T_i$ is the hypothetical natural period of vibration of the soil column based on considering wave reflections to be limited to the part of the soil column down to the mid-height of the $i$th soil layer; if $i$ is equal to $N$, then layer $i + 1$ refers to the bedrock.

(3) Employ the square-root-of-the-sum-of-the-squares (SRSS) combination rule to combine the modal contributions from the first two vibration modes for calculation of the effective shear strain of each soil layer.

$$\gamma_{eff,i} = 0.65 \times \sqrt{\gamma_{max,i,1}^2 + \gamma_{max,i,2}^2} \tag{30}$$

(4) Repeat steps (3)–(7) until the percentage differences of damping ratio ($\zeta_j$) and shear modulus reduction ratio ($G_{red,i}$) in two consecutive iterations are within a pre-defined limit of tolerance. The tolerance limit of 1% was adopted in the study.

With the shear strain profiles determined, the three parameters, the minimum reduced *SWV* $V_i$, the shifted first modal natural period $T_{site,1}$, and the averaged damping ratio $\zeta$, are to be determined from the last iteration by the use of Equations (19), (20), and (26). The sampling scheme identifies up to two critical soil columns with the period-dependent criteria as described in the following: $T_{structure}$ refers to the fundamental natural period of vibration of the structure, and $T_{site}$ refers to the initial site natural period as calculated using Equation (9).

- If $T_{structure} \leq 0.9 T_{site}$, two soil columns with the first having the **lowest minimum reduced *SWV*** and the second having the **lowest value of damping** are to be selected.

- If $T_{structure} > 0.9T_{site}$, two soil columns with the first having the **highest shifted period** and the second one having the **lowest value of damping** are to be selected.

Two soil column models are sampled, and both are simplified (in the manner as presented in Section 4.2) and processed for site response analysis (as presented in Section 4.3). The soil column that results in a higher amplification ratio at $T_{structure}$ is taken as the output parameter to be reported. The sampling process is illustrated by a case study to be presented in the latter part of the paper (Section 5).

### 4.2. Simplifying Soil Column Models

The code of practice specifies that borelogs should report soil properties for each soil layer with a thickness of up to 1.5 m [31]. A four-step procedure is introduced herein to reduce the number of soil layers in the critical soil columns without resulting in significant modelling errors.

(1) Construct the shear strain profile of the soil column using the iterative process presented in Section 4.1;
(2) From the constructed shear strain profile, identify the first high strain zone by locating the soil layer with the peak shear strain and the adjacent layers where the shear strain values are higher than 80% of the peak strain. Merge all layers in the first high strain zone and calculate the averaged shear wave velocity and density value for that zone;
(3) Identify the second high strain zone by identifying another soil layer which has the peak shear strain and is outside the first high strain zone. Merge all soil layers in the second high strain zone and calculate the averaged shear wave velocity and density value for that zone;
(4) Merge the remaining adjacent soil layers that are bound in between the two high strain zones, the bedrock, and the soil surface. Calculate the averaged shear wave velocity and density for each of the bounded zones.

Figure 5 illustrates a simplified soil column model based on the estimated shear strain profile for the selected input motion nos. 1–2, 7–10, and 13–20 in Table 6. The red dotted lines represent the strain limit corresponding to 80% of the first and second peak strain values. Soil layers in the high strain zones are highlighted by the red circular symbols.

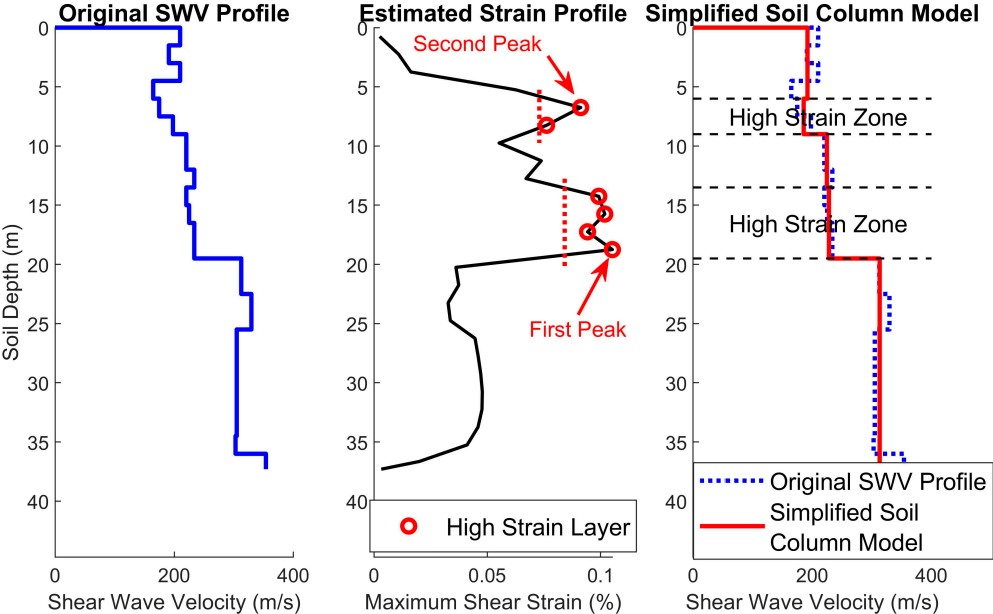

**Figure 5.** Simplified soil column model based on the estimated shear strain profile under bedrock motion nos. 1–2, 7–10, and 13–20.

### 4.3. Equivalent Linear Analysis and Accelerogram Processing

Many computer programs have been developed to operate the widely adopted one-dimensional equivalent linear analysis to simulate soil surface motions, such as SHAKE2000 [32], EERA [33], and Strata [34]. Routine 3 of the Quake Advice online program provides solutions to the equivalent linear analysis consistent with other commercial software as mentioned above. The Quake Advice program also features automatic baseline correction following the algorithm by Nigam and Jennings [35] to ensure that the velocity of the soil surface motion reaches zero at the end of each record.

Applying the procedures presented in Sections 4.1 and 4.2, the two simplified soil column models are sampled for site response analysis, giving two soil surface response spectra for each input bedrock motion. The one with a higher spectral amplitude at $T_{structure}$ is taken as the output site-specific response spectrum. The 24 sourced records for defining the input bedrock motions are processed to generate an ensemble of 24 site-specific response spectra. In addition, 12–16 corresponding soil surface accelerograms are highlighted for nonlinear time history analysis according to the selection scheme as illustrated in Figure 4.

## 5. Case Study

The execution of the three computational routines as presented in the paper are illustrated herein by a case study in support of the site-specific seismic design of two hypothetical built facilities in South Melbourne, Australia, for a 2500-year return period. The structures to be designed were reinforced concrete buildings with estimated fundamental periods of 0.5 and 1 s. Nine borelogs had been retrieved from site investigation with layer thickness and SPT blow count values as presented in Appendix C. The plasticity index was assumed to be 10% for low-plasticity clay. The input parameters are listed in Table 8.

**Table 8.** Input parameters into Routine 1–3 for the case study.

| Input Parameters [1] | Value | Unit |
|---|:---:|:---:|
| Routine 1 | | |
| Bedrock shear wave velocity | 800 | m/s |
| Energy ratio | 1 | |
| Shear wave velocity model | Imai and Tonouchi Model | |
| Material curve model | Hardin and Drnevich Model | |
| Dominant soil type | Low-plasticity clay | |
| Routine 2 | | |
| Return period | 2500 | year |
| Notional peak ground acceleration | 0.144 | g |
| Fundamental period of the structure | Structure no. 1: 0.5 Structure no. 2: 1.0 | second |

[1] The detailed borehole record data are presented in Appendix C.

Routine 1 presents the following output: the material curves, soil density profiles, and soil *SWV* profiles of nine borelogs. The construction site has an averaged site natural period of 0.614 s and is, therefore, classified as a $D_e$ site as per AS1170.4 R2018. Table 9 summarises the soil column properties and Figure 6 presents the *SWV* profiles, both demonstrating the diversity of details in the nine soil columns.

**Table 9.** Soil column properties of the nine borelogs.

| Soil Column No. | Total Thickness (m) | Initial Site Period (s) | Averaged Soil SWV (m/s) | Averaged Soil Density (kg/m³) |
|---|---|---|---|---|
| 1 | 37.3 | 0.603 | 247.6 | 1500 |
| 2 | 37.6 | 0.617 | 243.6 | 1500 |
| 3 | 37.3 | 0.610 | 244.7 | 1500 |
| 4 | 37.9 | 0.612 | 247.6 | 1500 |
| 5 | 37.7 | 0.620 | 243.3 | 1500 |
| 6 | 36.7 | 0.615 | 238.6 | 1500 |
| 7 | 37.8 | 0.619 | 244.2 | 1500 |
| 8 | 37.4 | 0.625 | 239.4 | 1500 |
| 9 | 37.4 | 0.608 | 246.1 | 1500 |

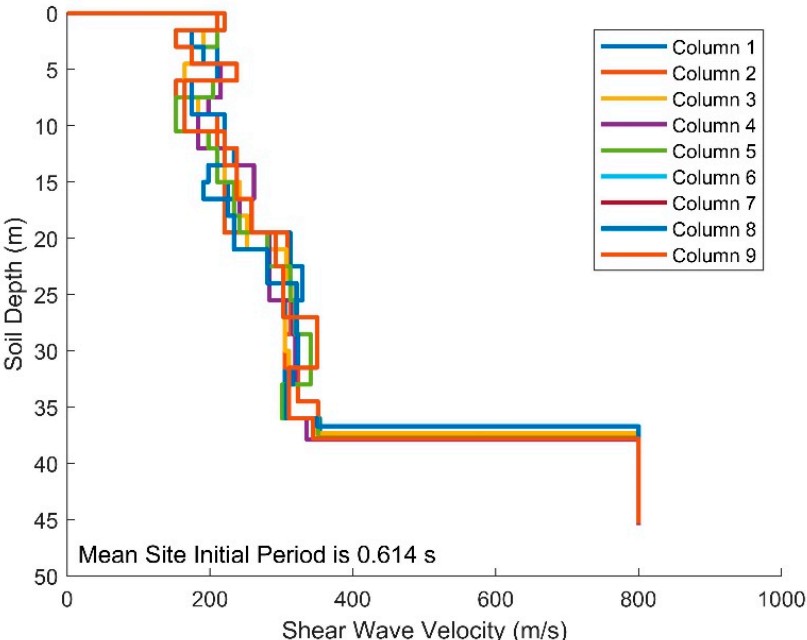

**Figure 6.** The shear wave velocity profiles of the nine borehole soil columns.

Four CMS were constructed in compliance with the Australian code spectrum model for rock (Class $B_e$) sites at the four distinctive periods: 0.2, 0.5, 1, and, 2 s. Six input motions at the bedrock level were sourced to match each of the CMS, totalling 24 input motions into site response analyses. Details of the earthquake events where the record was taken are summarised in Table 6. The response spectra of the selected and scaled input (bedrock) motions are presented in Figure 7. The bedrock motion data of the ensemble are available for downloading at https://quakeadvice.org.

The different estimated fundamental natural periods of the two case study structures resulted in two sets of bedrock motions to be employed for sampling the critical soil columns. With the selection scheme illustrated in Figure 4, motion nos. 1–2, 7–16, and 19–20 were selected for the case study structure no. 1 ($T_{structure} = 0.5$ s), whereas motion nos. 1–2, 7–10, and 13–20 were selected for the case study structure no. 2 ($T_{structure} = 1$ s).

Three dynamic soil properties were determined based on the sampling procedure described in Section 4.1. With case study structure no. 1 ($T_{structure} = 0.5$ s is smaller than 0.9 times the initial site period of $T_{site} = 0.614$ s), the 1st critical soil column was the one with the lowest minimum reduced SWV ($V_i$) whereas the 2nd critical soil column was the one with the lowest averaged damping ratio $\zeta$. By contrast, case study structure no. 2 had a higher natural period ($T_{structure} = 1$ s), which exceeded 0.9 times $T_{site}$. The corresponding critical soil columns were accordingly the one with the highest shifted first modal period $T_{site,1}$ and another one with the lowest averaged damping ratio $\zeta$. The computational

results of the parameters are summarised in Table 10, showing that soil column nos. 3 and 7 shall be sampled for case study structure no. 1, whereas soil column nos. 3 and 5 for case study structure no. 2. The sampled soil columns have been simplified based on the respective estimated shear strain profiles (Figure 8). Although soil column no. 3 was selected for both case study structures, descriptions of the soil layers in the soil column were simplified differently, given the difference with the estimated shear strain profiles.

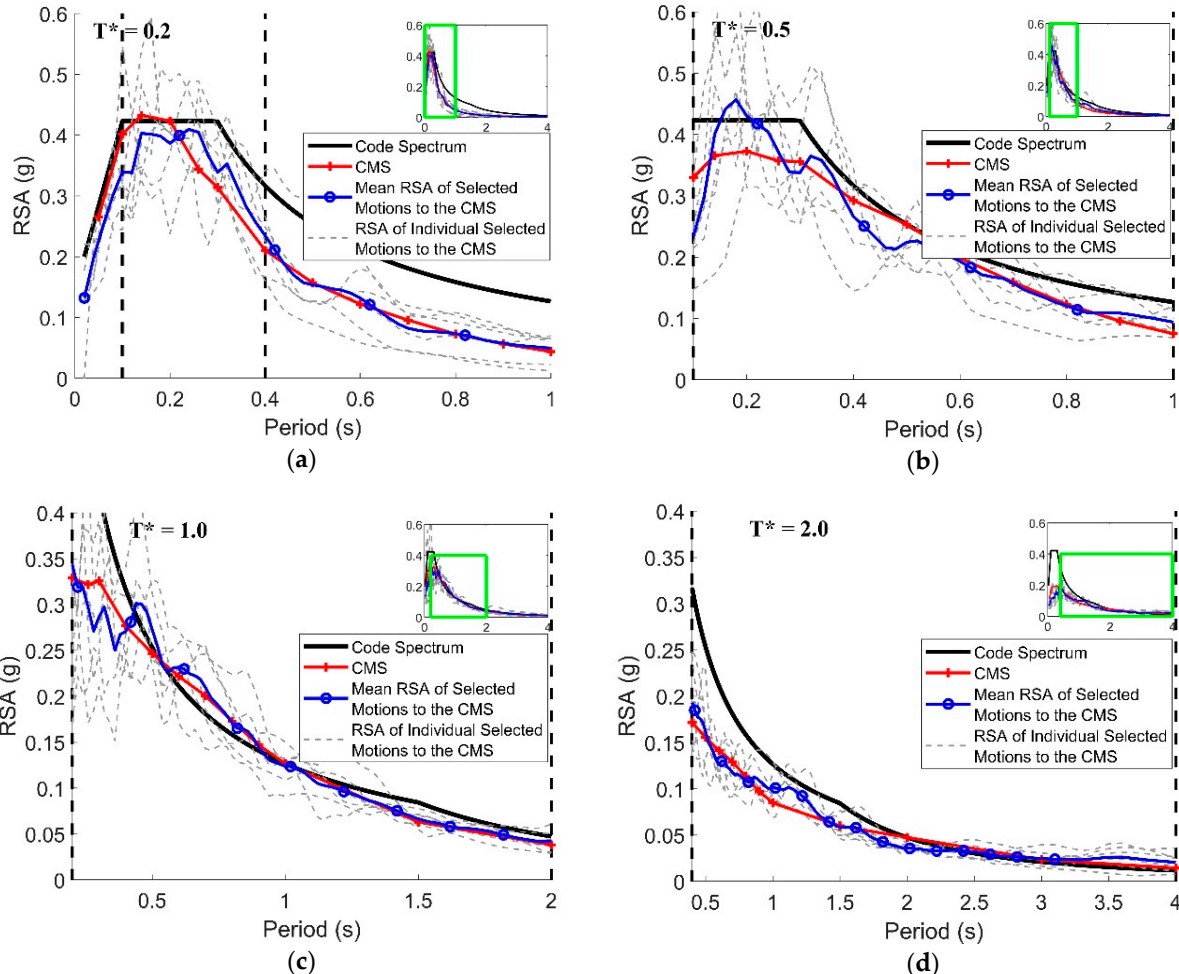

**Figure 7.** The response spectra of the selected and scaled motions for defining bedrock excitation at the four reference periods: (**a**) $T^* = 0.2$ s; (**b**) $T^* = 0.5$ s; (**c**) $T^* = 1$ s; (**d**) $T^* = 2$ s. The part of the figure in the green square is enlarged to show details.

**Table 10.** Soil parameters to identify the critical soil columns.

| Soil Column No. | $T_{structure}$ = 0.5 s | | $T_{structure}$ = 1 s | |
|---|---|---|---|---|
| | Minimum $V_i$ (m/s) | $\zeta$ | $T_{site,1}$ (s) | $\zeta$ |
| 1 | 109.5 | 9.46 | 0.851 | 9.90 |
| 2 | 75.6 | 9.88 | 0.896 | 10.23 |
| 3 | 108.8 | 9.41 | 0.863 | 9.80 |
| 4 | 85.7 | 9.62 | 0.867 | 10.03 |
| 5 | 66.2 | 10.20 | 0.919 | 10.63 |
| 6 | 86.3 | 10.15 | 0.885 | 10.66 |
| 7 | 56.8 | 10.45 | 0.911 | 10.81 |
| 8 | 65.7 | 10.43 | 0.917 | 10.84 |
| 9 | 64.8 | 10.36 | 0.889 | 10.87 |

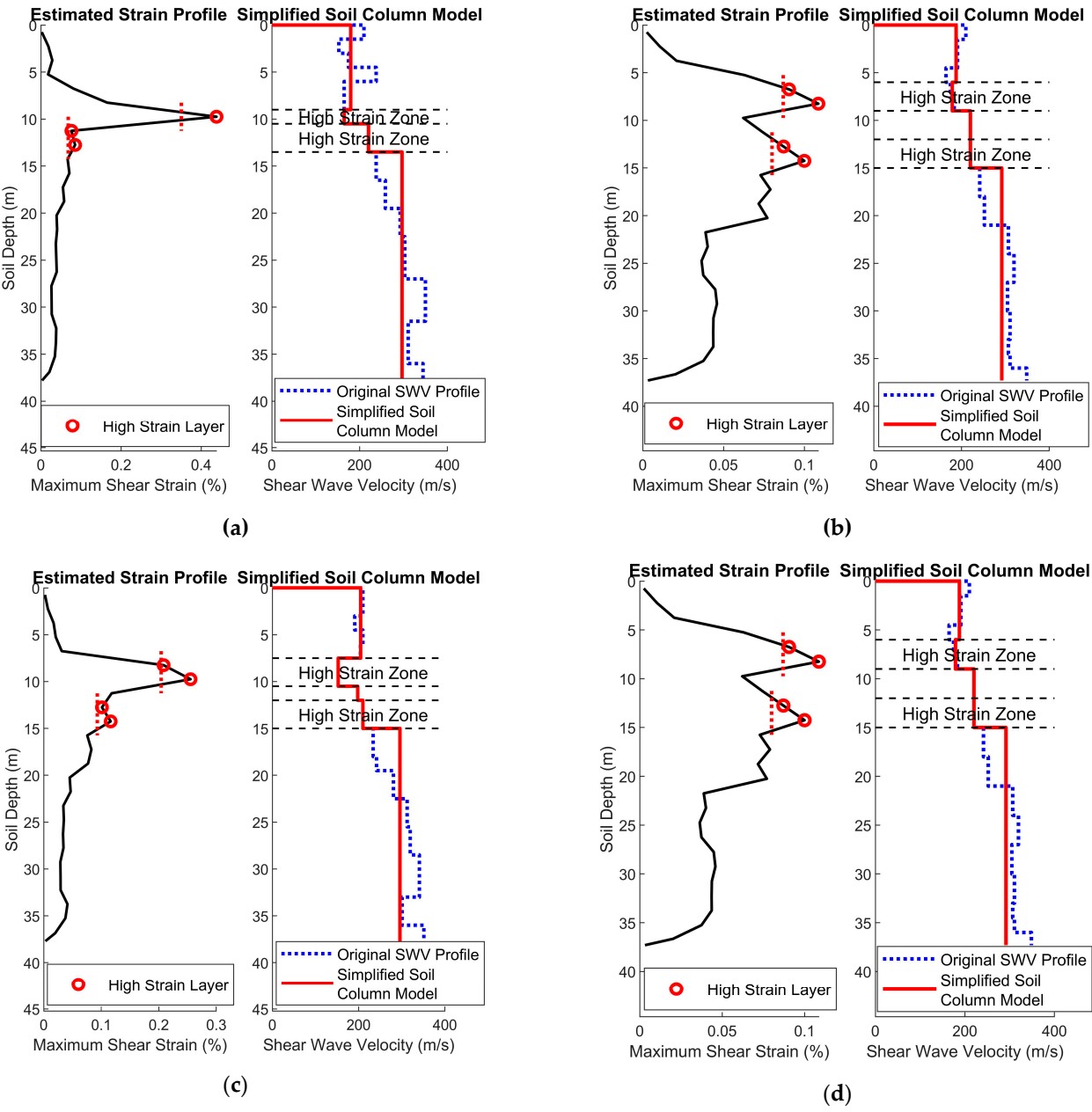

**Figure 8.** Simplified soil column model for the case study: (**a**) soil column no. 7 for structure no. 1; (**b**) soil column no. 3 for structure no. 1; (**c**) soil column no. 5 for structure no. 2; (**d**) soil column no. 3 for structure no. 2.

With each case study structure, Routine 3 was then applied to the two simplified soil column models that were subject to equivalent linear analysis using the 24 input bedrock motions. Two soil surface response spectra were generated for each input motion, and the one showing more conservative predictions was taken and included into calculations for the ensemble of site-specific response spectra. This part of the procedure is demonstrated by processing input motion no. 1 using the two simplified soil column models for case study structure no. 1. As shown in Figure 9, the soil surface response spectrum as derived from analysis of soil column no. 7 had a higher amplitude at $T_{structure} = 0.5$ s and was, therefore, included into the ensemble of soil surface response spectra as output from the procedure. Twenty-four site-specific response spectra were determined for each of the two case study structures as presented in Figures 10 and 11, respectively.

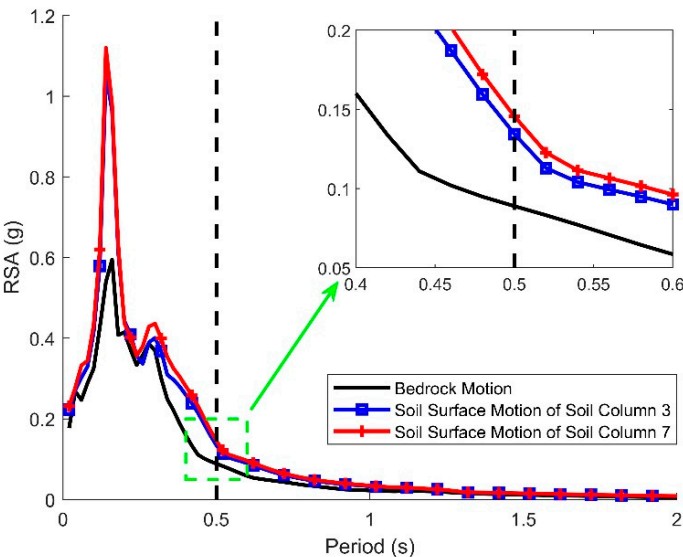

**Figure 9.** Soil surface response spectra of the simplified soil column model nos. 3 and 7 for structure no. 1 when subject to bedrock motion no. 1. The part of the figure in the green square is enlarged to show details.

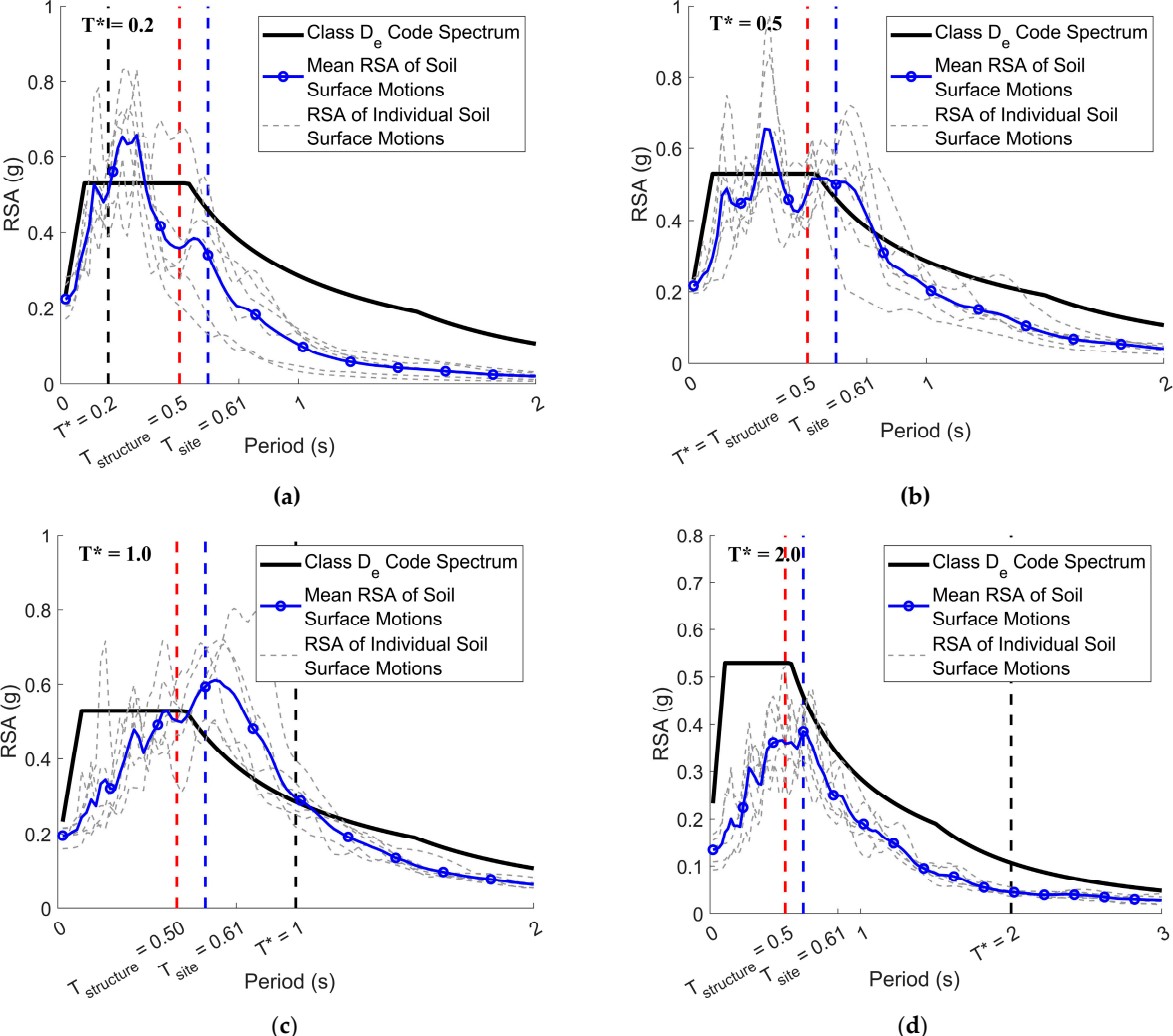

**Figure 10.** The site-specific response spectra at the four reference periods as output for structure no. 1 for the case study: (**a**) $T^* = 0.2$ s; (**b**) $T^* = 0.5$ s; (**c**) $T^* = 1$ s; (**d**) $T^* = 2$ s.

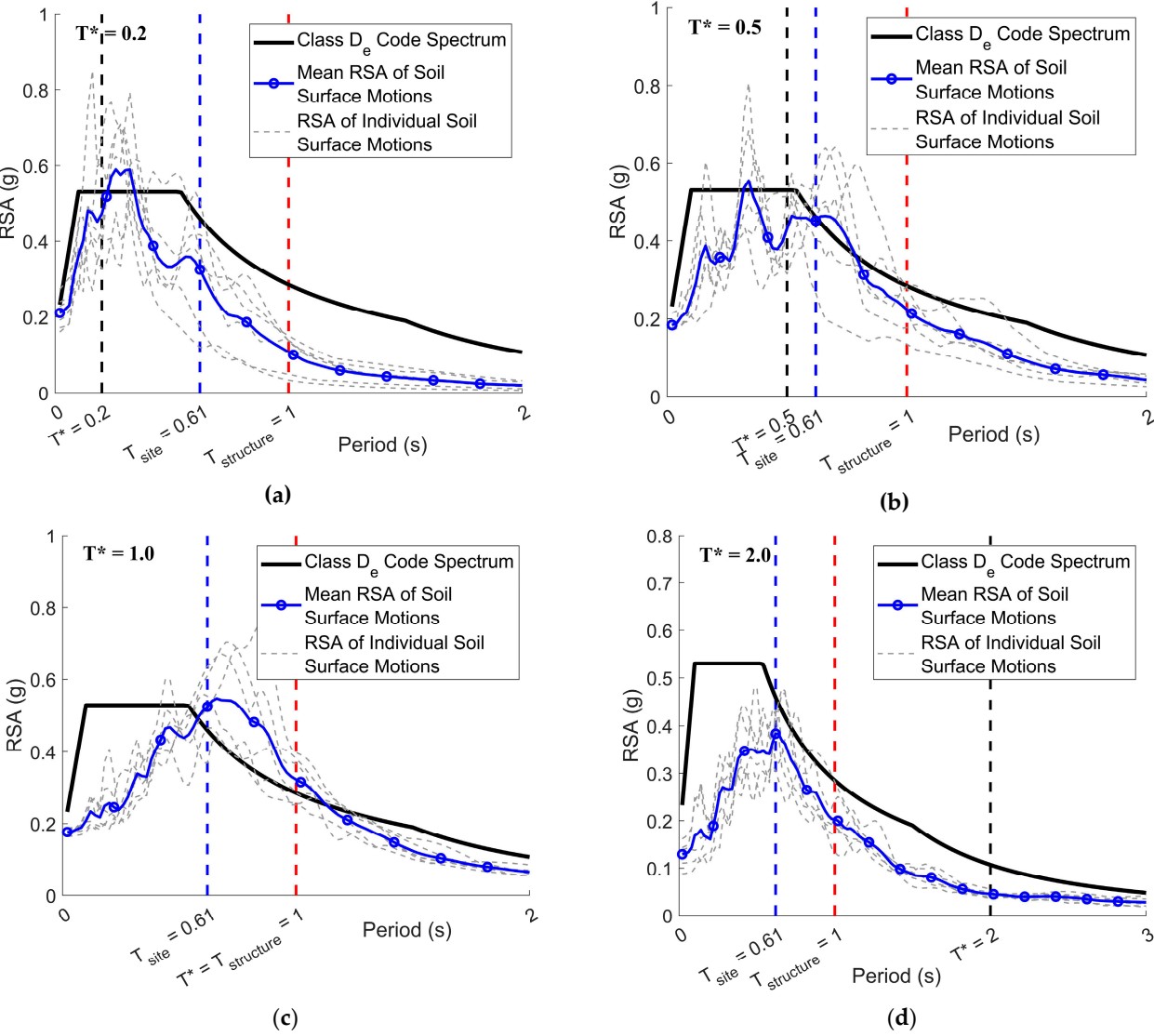

**Figure 11.** The site-specific response spectra at the four reference periods as output for structure no. 2 for the case study: (**a**) $T^* = 0.2$ s; (**b**) $T^* = 0.5$ s; (**c**) $T^* = 1$ s; (**d**) $T^* = 2$ s.

In addition to generating the 24 site-specific response spectra, Routine 3 also served the purpose of generating soil surface accelerograms for nonlinear time history analysis of the case study building structures. The authors recommend using motion nos. 1–2, 7–16, and 19–20 for case study structure no. 1; and motion nos. 1–2, 7–10, and 13–20 for case study structure no. 2. The acceleration time-histories of the accelerogram records were plotted in Appendix D for reference.

## 6. Closing Remarks

This paper aims to generate site-specific soil surface response spectra involving (1) analyses of multiple borelogs taken from the targeted site, (2) use of the CMS methodology for generating scenario-specific input ground motions at the bedrock level, and (3) site response analyses of critical (sampled) soil column models that have been simplified. The presented procedure, which is freely accessible at https://quakeadvice.org (accessed on 1 December 2022), has many advantages over the code method. Firstly, the use of site response analyses of soil column models gives more accurate predictions than the code response spectrum models that are based on broad site classifications. The CMS methodology for sampling and scaling accelerograms sourced from the NGA-West2 database has the added benefit of

providing more realistic representations of input motions at the bedrock level than simply scaling accelerograms to match with the code-stipulated spectrum. Meanwhile, the site information sampling method to identify and simplify the critical soil column models serves to reduce computational time. Detailed step-by-step descriptions of the procedure complete with data tables and the use of a case study to assist with the illustration are presented in this paper to guide practical applications of the methodology. Note, 1D site response (soil column) analysis as presented in this article may only be applied at locations where there are neither major 2D, nor 3D, stratigraphical features in the sedimentary layers. There are also uncertainties associated with the way dynamic properties of the soil materials are modelled and *SWV* are calculated. Recognising these limitations and uncertainties, designers would need to repeat analyses of the soil columns, whilst varying input parameters to track sensitivities in order to achieve a robust modelling outcome. The sampling methodology presented in this article serves to facilitate this approach to modelling through fast-tracking soil column analyses.

**Author Contributions:** Conceptualization, N.L., H.-H.T.; methodology, Y.H., N.L.; software, Y.H., P.K.; resources, S.M.; writing—original draft preparation, Y.H., N.L.; writing—review and editing, P.K., H.-H.T., S.M.; supervision, N.L. All authors have read and agreed to the published version of the manuscript.

**Funding:** This research received financial support from the Australian Research Council (ARC) Discovery Project DP180101593 entitled "Seismic Performance of Precast Concrete Buildings for Lower Seismic Regions".

**Data Availability Statement:** The data presented in this study are on open access at https://quakeadvice.org, accessed on 1 December 2022.

**Conflicts of Interest:** The authors declare no conflict of interest.

### Nomenclature

| | | | |
|---|---|---|---|
| $G_{max}$ | maximum shear modulus | $\gamma_{eff}$ | effective shear strain |
| $G_{red}$ | shear modulus reduction ratio | $\gamma_{max}$ | maximum shear strain |
| $G_{sec}$ | secant shear modulus | $\gamma_{ref}$ | reference shear strain |
| $H_i$ | thickness of layer i | $\zeta$ | weighted average of damping ratio of soil layers |
| i | layer number in a soil column | $\zeta_{initial}$ | initial damping ratio |
| N | total number of soil layers | $\zeta_{max}$ | maximum damping ratio |
| $N_{measured}$ | measured SPT blow count | $\varepsilon(T^*)$ | epsilon at $T^*$, defined as the number of standard deviations required to match the target spectral amplitude at $T^*$ |
| $N_{60}$ | normalised SPT blow count | $\mu(T)$ | estimated median response spectral acceleration at period T |
| $R_{jb}$ | Joyner-Boore distance | $\rho_R$ | density of bedrock |
| Sa | target response spectral acceleration | $\rho_S$ | weighted average of density of soil layers |
| T | period | $\rho(T, T^*)$ | correlation coefficient at period $T$ with reference to period $T^*$ |
| $T_i$ | hypothetical natural period down to the mid-height of layer i | $\sigma(T)$ | estimated standard deviation at period T |
| $T_{site}$ | initial site natural period | CMS | conditional mean spectra |
| $T_{site,1}$ | shifted first mode natural period of the site | GMPEs | ground motion prediction expressions |
| $T_{site,2}$ | shifted second mode natural period of the site | M-R | magnitude—distance |
| $T_{structure}$ | fundamental period of vibration of the structure | MSE | mean-squared error |
| $T^*$ | reference period | PEER | Pacific Earthquake Engineering Research Centre |
| $V_i$ | reduced shear wave velocity of layer i | PI | plasticity index |
| $V_R$ | shear wave velocity of bedrock | RSV | response spectral velocity |
| $V_S$ | time-weighted average of shear wave velocity of soil layers | SPT | standard penetration test |
| $\alpha$ | impedance ratio | SRSS | square root of the sum of the squares |
| $\gamma$ | shear strain | SWV | shear wave velocity |

## Appendix A  Soil Density

Appendix A presents recommended soil densities values (in kg/m³) based on plasticity and water content for cohesive soil in Table A1(a,b); and relative density, major division, and water content for cohesionless soils in Table A1(a,b). The terms and initials are explained in Table A2(a–c). Both Table A1(a,b) and Table A2(a–c) have been previously published in [8].

**Table A1.** Soil density estimated from detailed soil type and moisture content: (**a**) soil density for cohesive soils; (**b**) soil density for cohesionless soil.

| (a) | | | | | |
|---|---|---|---|---|---|
| **Soil Type** | **Water Content** | | | | |
| | **M1** | **M2** | **M3** | **W1** | **W2** |
| ML | 1490 | 1590 | 1600 | 1590 | 1600 |
| MH | 1590 | 1610 | 1650 | 1730 | 1720 |
| CL | 1410 | 1440 | 1490 | 1610 | 1540 |
| CI | 1440 | 1510 | 1600 | 1610 | 1640 |
| CH | 1600 | 1720 | 1540 | 1640 | 1720 |

| (b) | | | | | | | | | | | | | | |
|---|---|---|---|---|---|---|---|---|---|---|---|---|---|---|
| **Soil Type** | **Water Content** | | | | | | | | | | | | | |
| | **Dry** | | | | | **Moist** | | | | | **Wet** | | | | |
| | **VL** | **L** | **MD** | **D** | **VD** | **VL** | **L** | **MD** | **D** | **VD** | **VL** | **L** | **MD** | **D** | **VD** |
| GP | 1790 | 1830 | 1910 | 1990 | 2040 | 1950 | 1990 | 2050 | 2120 | 2160 | 2110 | 2140 | 2190 | 2240 | 2270 |
| GW | 1800 | 1860 | 1970 | 2100 | 2180 | 1960 | 2010 | 2100 | 2200 | 2270 | 2120 | 2160 | 2230 | 2310 | 2360 |
| GM | 1640 | 1700 | 1810 | 1940 | 2020 | 1830 | 1880 | 1970 | 2070 | 2140 | 2020 | 2060 | 2130 | 2210 | 2260 |
| GC | 1640 | 1700 | 1810 | 1940 | 2020 | 1830 | 1880 | 1970 | 2070 | 2140 | 2020 | 2060 | 2130 | 2210 | 2260 |
| SP | 1560 | 1620 | 1730 | 1860 | 1940 | 1760 | 1810 | 1900 | 2010 | 2070 | 1970 | 2010 | 2080 | 2160 | 2210 |
| SW | 1570 | 1640 | 1790 | 1960 | 2070 | 1770 | 1830 | 1950 | 2090 | 2180 | 1980 | 2020 | 2110 | 2220 | 2290 |
| SM | 1340 | 1430 | 1610 | 1840 | 2020 | 1580 | 1660 | 1810 | 2000 | 2140 | 1830 | 1890 | 2000 | 2150 | 2260 |
| SC | 1410 | 1490 | 1650 | 1840 | 1980 | 1640 | 1710 | 1840 | 1990 | 2100 | 1880 | 1930 | 2020 | 2140 | 2230 |

**Table A2.** Explanation of soil terminologies: (**a**) initials for soil type; (**b**) initials for water content; (**c**) soil descriptions and correlation to SPT count.

| (a) | | | |
|---|---|---|---|
| **Cohesive Soil** | | **Cohesionless Soil** | |
| **Initial** | **Soil Type** | **Initial** | **Soil Type** |
| ML | Low-plasticity silt | GW | Well-grade gravel |
| MH | High-plasticity silt | GP | Poorly-grade gravel |
| CL | Low-plasticity clay | GM | Silty gravel |
| CI | Medium-plasticity clay | GC | Clayey gravel |
| CH | High-plasticity clay | SW | Well-grade sand |
| | | SP | Poorly-grade sand |
| | | SM | Silty sand |
| | | SC | Clayey sand |

**Table A2.** *Cont.*

| (b) | | | | | |
|---|---|---|---|---|---|
| **Cohesive Soil** | | | **Cohesionless Soil** | | |
| **Initial** | **Term** | **Water Content** | **Initial** | **Water Content** | |
| M1 | $w < PL$ | Moist, dry of plastic limit | D | Dry | |
| M2 | $w \approx PL$ | Moist, near plastic limit | M | Moist | |
| M3 | $w > PL$ | Moist, wet of plastic limit | W | Wet | |
| W1 | $w \approx LL$ | Wet, near liquid limit | | | |
| W2 | $w > LL$ | Wet, wet of liquid limit | | | |

| (c) | | | | | |
|---|---|---|---|---|---|
| **Cohesive Soil** | | | **Cohesionless Soil** | | |
| **Initial** | **Description** | **SPT Count** | **Initial** | **Description** | **SPT Count** |
| VS | Very soft | 0–2 | VL | Very loose | 0–4 |
| S | Soft | 2–4 | L | Loose | 4–10 |
| F | Firm | 4–8 | MD | Medium dense | 10–30 |
| St | Stiff | 8–15 | D | Dense | 30–50 |
| VSt | Very stiff | 15–30 | VD | Very dense | >50 |
| H | Hard | >30 | | | |

## Appendix B  Details of the Example Borehole Record

Table A3 contains details of the example borehole record to supplement Figure 1. The borehole record was retrieved from the site investigation report for a construction project in North Melbourne, VIC, Australia.

**Table A3.** Borehole record and the estimated *SWV* and density profile.

| **Input Borehole Information** | | | | **SWV and Density Profile** | |
|---|---|---|---|---|---|
| **Layer No.** | **Thickness (m)** | **$N_{60}$** | **Soil Type** | **SWV (m/s)** | **Density (kg/m$^3$)** |
| 1 | 1.5 | 10 | Low-plasticity clay | 210 | 1500 |
| 2 | 1.5 | 7 | Low-plasticity clay | 191 | 1500 |
| 3 | 1.5 | 10 | Low-plasticity clay | 210 | 1500 |
| 4 | 1.5 | 3 | Low-plasticity clay | 153 | 1500 |
| 5 | 1.5 | 3 | Low-plasticity clay | 153 | 1500 |
| 6 | 1.5 | 8 | Low-plasticity clay | 198 | 1500 |
| 7 | 1.5 | 12 | Low-plasticity clay | 220 | 1500 |
| 8 | 1.5 | 12 | Low-plasticity clay | 220 | 1500 |
| 9 | 1.5 | 15 | Low-plasticity clay | 234 | 1500 |
| 10 | 1.5 | 12 | Low-plasticity clay | 220 | 1500 |
| 11 | 1.5 | 13 | Low-plasticity clay | 225 | 1500 |
| 12 | 1.5 | 15 | Low-plasticity clay | 234 | 1500 |
| 13 | 1.5 | 15 | Low-plasticity clay | 234 | 1500 |
| 14 | 1.5 | 45 | Low-plasticity clay | 312 | 1500 |
| 15 | 1.5 | 45 | Low-plasticity clay | 312 | 1500 |

**Table A3.** *Cont*.

| Input Borehole Information | | | | SWV and Density Profile | |
|---|---|---|---|---|---|
| Layer No. | Thickness (m) | $N_{60}$ | Soil Type | SWV (m/s) | Density (kg/m³) |
| 16 | 1.5 | 55 | Low-plasticity clay | 329 | 1500 |
| 17 | 1.5 | 55 | Low-plasticity clay | 329 | 1500 |
| 18 | 1.5 | 41 | Low-plasticity clay | 305 | 1500 |
| 19 | 1.5 | 41 | Low-plasticity clay | 305 | 1500 |
| 20 | 1.5 | 41 | Low-plasticity clay | 305 | 1500 |
| 21 | 1.5 | 41 | Low-plasticity clay | 305 | 1500 |
| 22 | 1.5 | 41 | Low-plasticity clay | 305 | 1500 |
| 23 | 1.5 | 41 | Low-plasticity clay | 305 | 1500 |
| 24 | 1.5 | 40 | Low-plasticity clay | 303 | 1500 |
| 25 | 1.3 | 72 | Low-plasticity clay | 354 | 1500 |
| Bedrock | - | - | *SWV* = 800 m/s | 800 | 2025 |

**Appendix C  Borehole Record Data for the Case Study**

Table A4 summarises the layer thicknesses (*H*) and SPT blow counts ($N_{60}$) for nine borehole records (i.e., BH1–BH9) retrieved from the case study construction site.

**Table A4.** Borehole record data for the case study.

| Layer No. | BH1 | | BH2 | | BH3 | | BH4 | | BH5 | | BH6 | | BH7 | | BH8 | | BH9 | |
|---|---|---|---|---|---|---|---|---|---|---|---|---|---|---|---|---|---|---|
| | *H* | $N_{60}$ | *H* | $N_{60}$ | *H* | $N_{60}$ | *H* | $N_{60}$ | *H* | $N_{60}$ | *H* | $N_{60}$ | *H* | $N_{60}$ | *H* | $N_{60}$ | *H* | $N_{60}$ |
| 1 | 1.5 | 10 | 1.5 | 12 | 1.5 | 10 | 1.5 | 10 | 1.5 | 10 | 1.5 | 10 | 1.5 | 10 | 1.5 | 5 | 1.5 | 12 |
| 2 | 1.5 | 7 | 1.5 | 7 | 1.5 | 7 | 1.5 | 5 | 1.5 | 10 | 1.5 | 5 | 1.5 | 3 | 1.5 | 4 | 1.5 | 5 |
| 3 | 1.5 | 10 | 1.5 | 7 | 1.5 | 7 | 1.5 | 5 | 1.5 | 7 | 1.5 | 7 | 1.5 | 5 | 1.5 | 4 | 1.5 | 5 |
| 4 | 1.5 | 4 | 1.5 | 4 | 1.5 | 4 | 1.5 | 11 | 1.5 | 10 | 1.5 | 10 | 1.5 | 16 | 1.5 | 10 | 1.5 | 10 |
| 5 | 1.5 | 5 | 1.5 | 3 | 1.5 | 5 | 1.5 | 11 | 1.5 | 9 | 1.5 | 5 | 1.5 | 4 | 1.5 | 4 | 1.5 | 6 |
| 6 | 1.5 | 8 | 1.5 | 5 | 1.5 | 6 | 1.5 | 8 | 1.5 | 3 | 1.5 | 5 | 1.5 | 4 | 1.5 | 4 | 1.5 | 6 |
| 7 | 1.5 | 12 | 1.5 | 10 | 1.5 | 12 | 1.5 | 6 | 1.5 | 3 | 1.5 | 12 | 1.5 | 4 | 1.5 | 6 | 1.5 | 4 |
| 8 | 1.5 | 12 | 1.5 | 10 | 1.5 | 12 | 1.5 | 6 | 1.5 | 8 | 1.5 | 12 | 1.5 | 12 | 1.5 | 5 | 1.5 | 5 |
| 9 | 1.5 | 15 | 1.5 | 16 | 1.5 | 12 | 1.5 | 12 | 1.5 | 10 | 1.5 | 12 | 1.5 | 12 | 1.5 | 16 | 1.5 | 20 |
| 10 | 1.5 | 12 | 1.5 | 16 | 1.5 | 12 | 1.5 | 23 | 1.5 | 10 | 1.5 | 8 | 1.5 | 16 | 1.5 | 18 | 1.5 | 16 |
| 11 | 1.5 | 13 | 1.5 | 12 | 1.5 | 17 | 1.5 | 23 | 1.5 | 15 | 1.5 | 7 | 1.5 | 16 | 1.5 | 18 | 1.5 | 16 |
| 12 | 1.5 | 15 | 1.5 | 12 | 1.5 | 17 | 1.5 | 17 | 1.5 | 15 | 1.5 | 13 | 1.5 | 22 | 1.5 | 18 | 1.5 | 18 |
| 13 | 1.5 | 15 | 1.5 | 12 | 1.5 | 20 | 1.5 | 17 | 1.5 | 17 | 1.5 | 15 | 1.5 | 22 | 1.5 | 32 | 1.5 | 18 |
| 14 | 1.5 | 45 | 1.5 | 43 | 1.5 | 20 | 1.5 | 31 | 1.5 | 30 | 1.5 | 15 | 1.5 | 35 | 1.5 | 32 | 1.5 | 43 |
| 15 | 1.5 | 45 | 1.5 | 43 | 1.5 | 42 | 1.5 | 31 | 1.5 | 30 | 1.5 | 30 | 1.5 | 35 | 1.5 | 35 | 1.5 | 43 |
| 16 | 1.5 | 55 | 1.5 | 43 | 1.5 | 42 | 1.5 | 31 | 1.5 | 45 | 1.5 | 30 | 1.5 | 40 | 1.5 | 35 | 1.5 | 43 |
| 17 | 1.5 | 55 | 1.5 | 45 | 1.5 | 49 | 1.5 | 31 | 1.5 | 45 | 1.5 | 50 | 1.5 | 40 | 1.5 | 22 | 1.5 | 43 |
| 18 | 1.5 | 41 | 1.5 | 45 | 1.5 | 49 | 1.5 | 47 | 1.5 | 49 | 1.5 | 50 | 1.5 | 40 | 1.5 | 48 | 1.5 | 47 |
| 19 | 1.5 | 41 | 1.5 | 45 | 1.5 | 41 | 1.5 | 47 | 1.5 | 49 | 1.5 | 50 | 1.5 | 69 | 1.5 | 53 | 1.5 | 47 |
| 20 | 1.5 | 41 | 1.5 | 41 | 1.5 | 41 | 1.5 | 49 | 1.5 | 62.5 | 1.5 | 51 | 1.5 | 69 | 1.5 | 53 | 1.5 | 65 |
| 21 | 1.5 | 41 | 1.5 | 41 | 1.5 | 44 | 1.5 | 49 | 1.5 | 62.5 | 1.5 | 51 | 1.5 | 69 | 1.5 | 57 | 1.5 | 50 |
| 22 | 1.5 | 41 | 1.5 | 51 | 1.5 | 44 | 1.5 | 49 | 1.5 | 62.5 | 1.5 | 47 | 1.5 | 44 | 1.5 | 57 | 1.5 | 50 |
| 23 | 1.5 | 41 | 1.5 | 51 | 1.5 | 42 | 1.5 | 43 | 1.5 | 39 | 1.5 | 42 | 1.5 | 44 | 1.5 | 57 | 1.5 | 50 |
| 24 | 1.5 | 40 | 1.5 | 70 | 1.5 | 44 | 1.5 | 43 | 1.5 | 39 | 1.5 | 42 | 1.5 | 44 | 1.5 | 57 | 1.5 | 52 |
| 25 | 1.3 | 72 | 1.6 | 70 | 1.3 | 68 | 1.9 | 59 | 1.7 | 70 | 0.7 | 69 | 1.8 | 65 | 1.4 | 65 | 1.4 | 65 |

## Appendix D  Soil Surface Accelerograms for the Case Study

Figures A1 and A2 present the soil surface accelerograms that are recommended for use in nonlinear time history analysis for structure nos. 1 and 2, respectively.

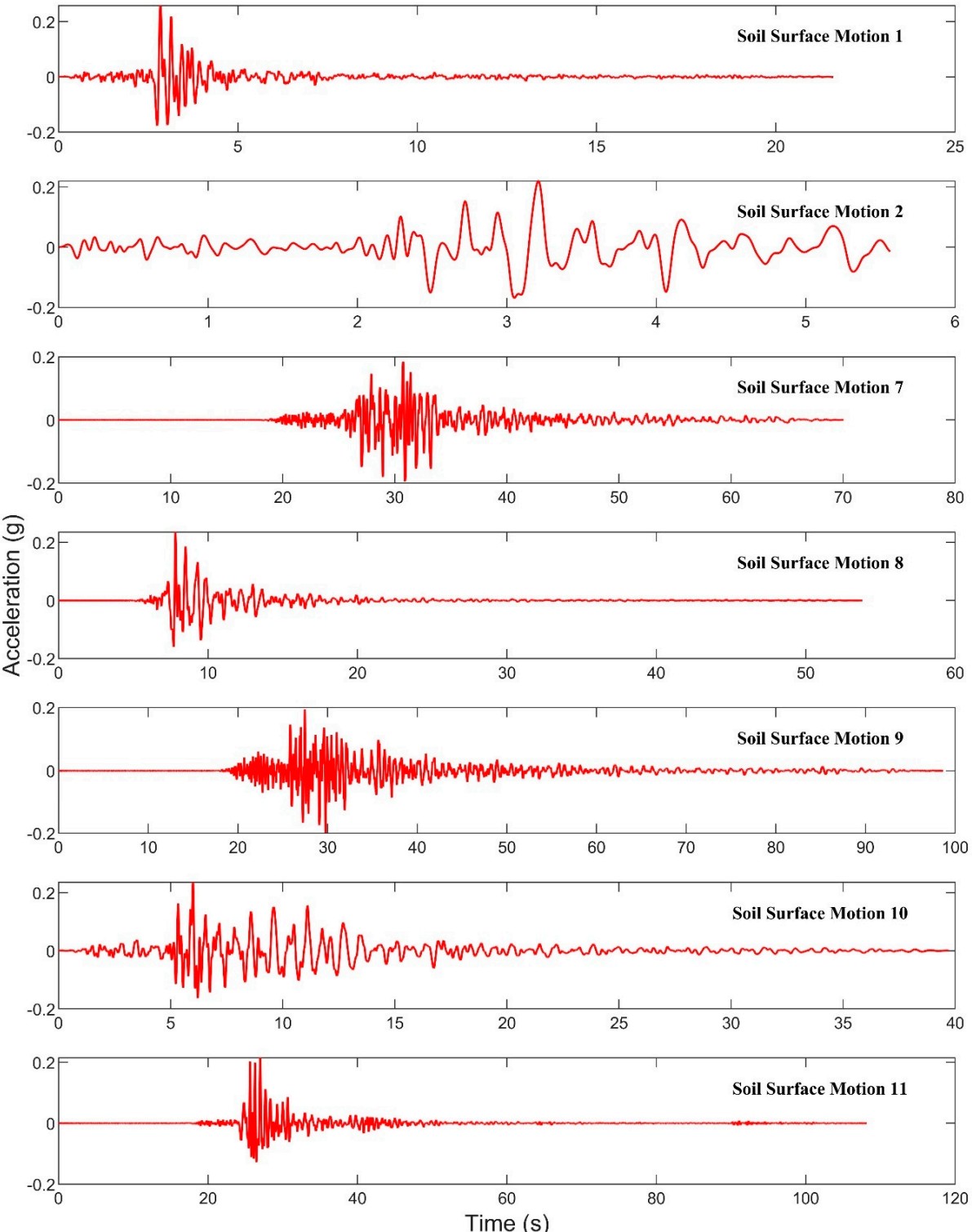

**Figure A1.** *Cont.*

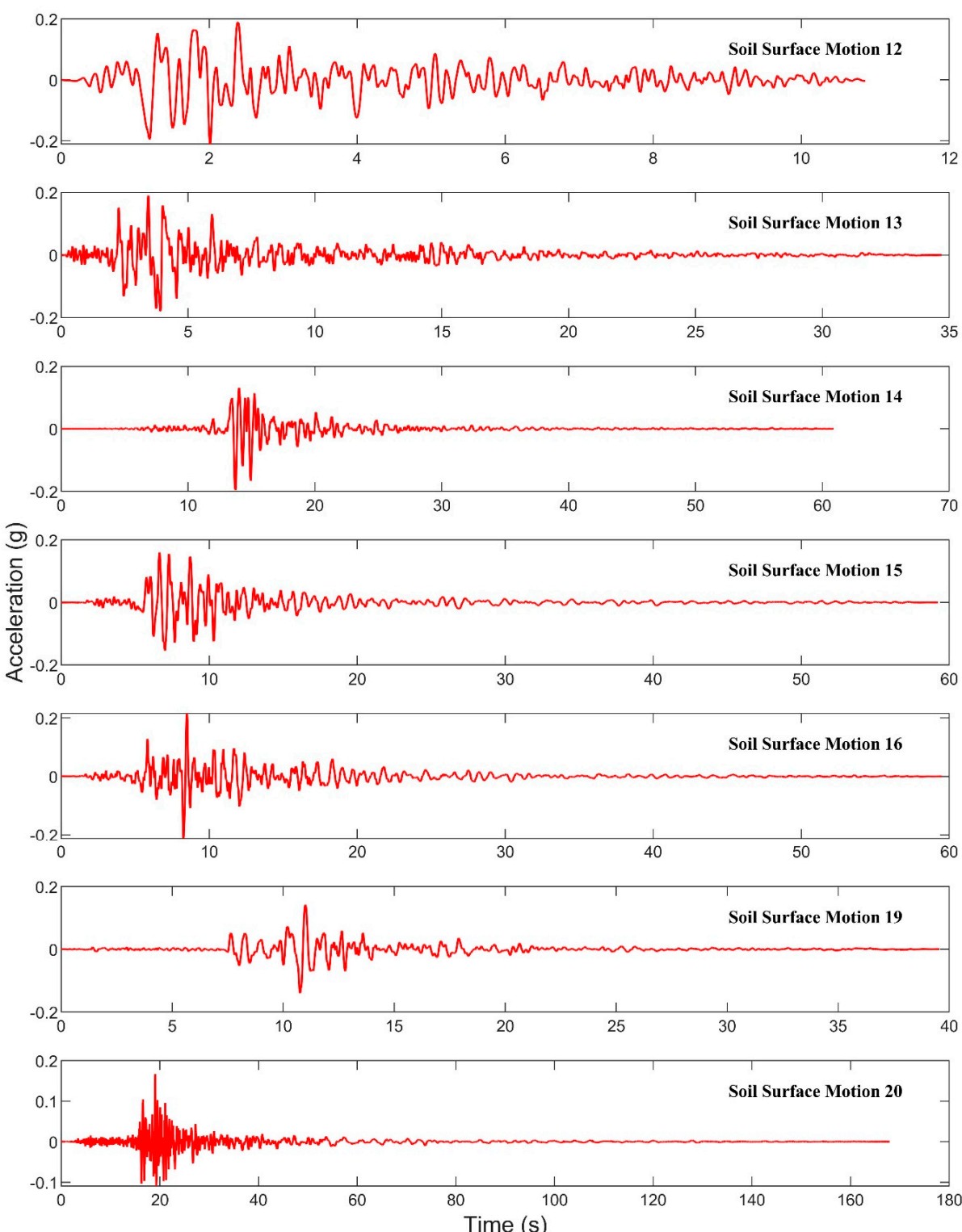

**Figure A1.** Soil surface accelerograms that are recommended for use in nonlinear time history analysis for structure no. 1 for the case study.

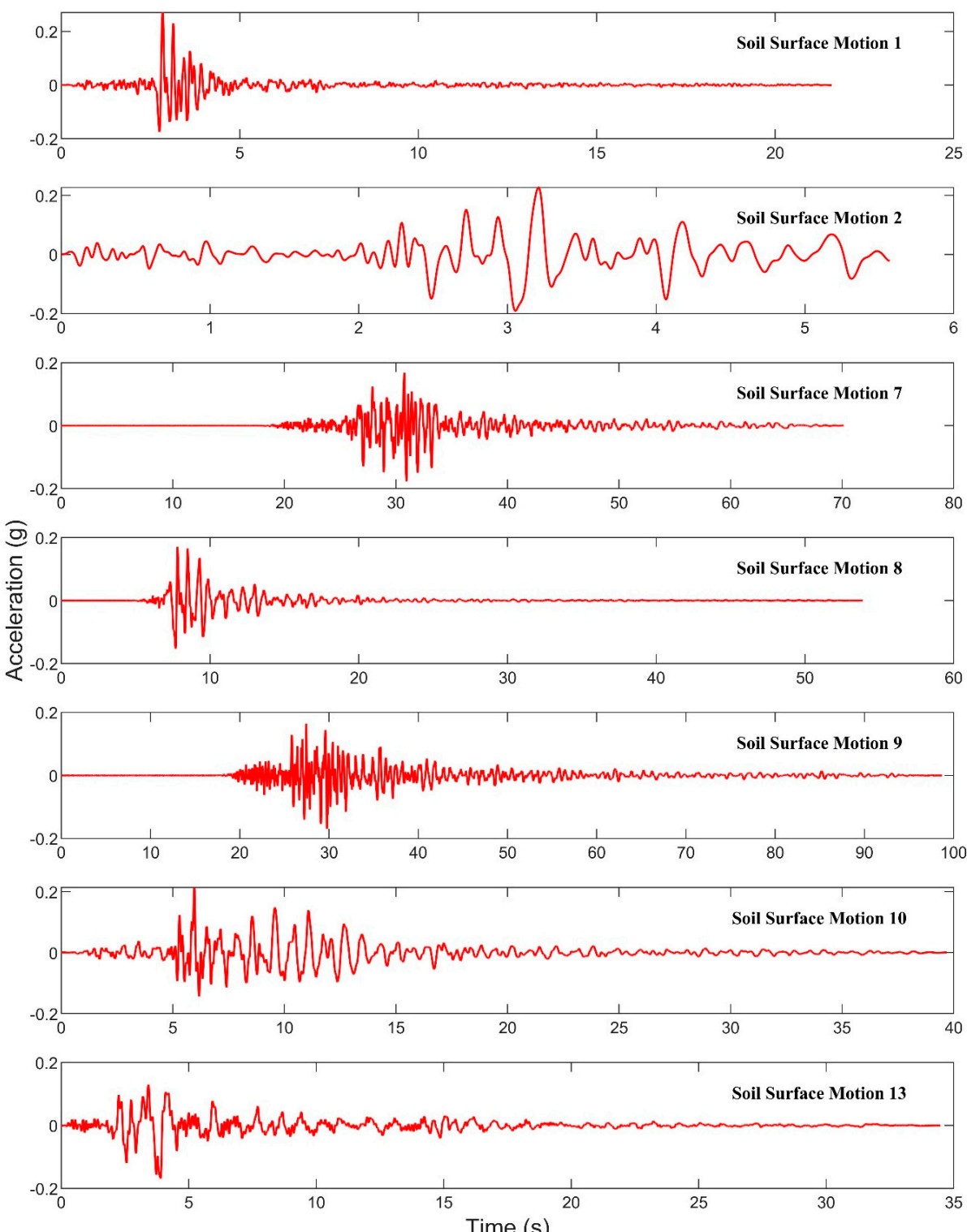

**Figure A2.** *Cont.*

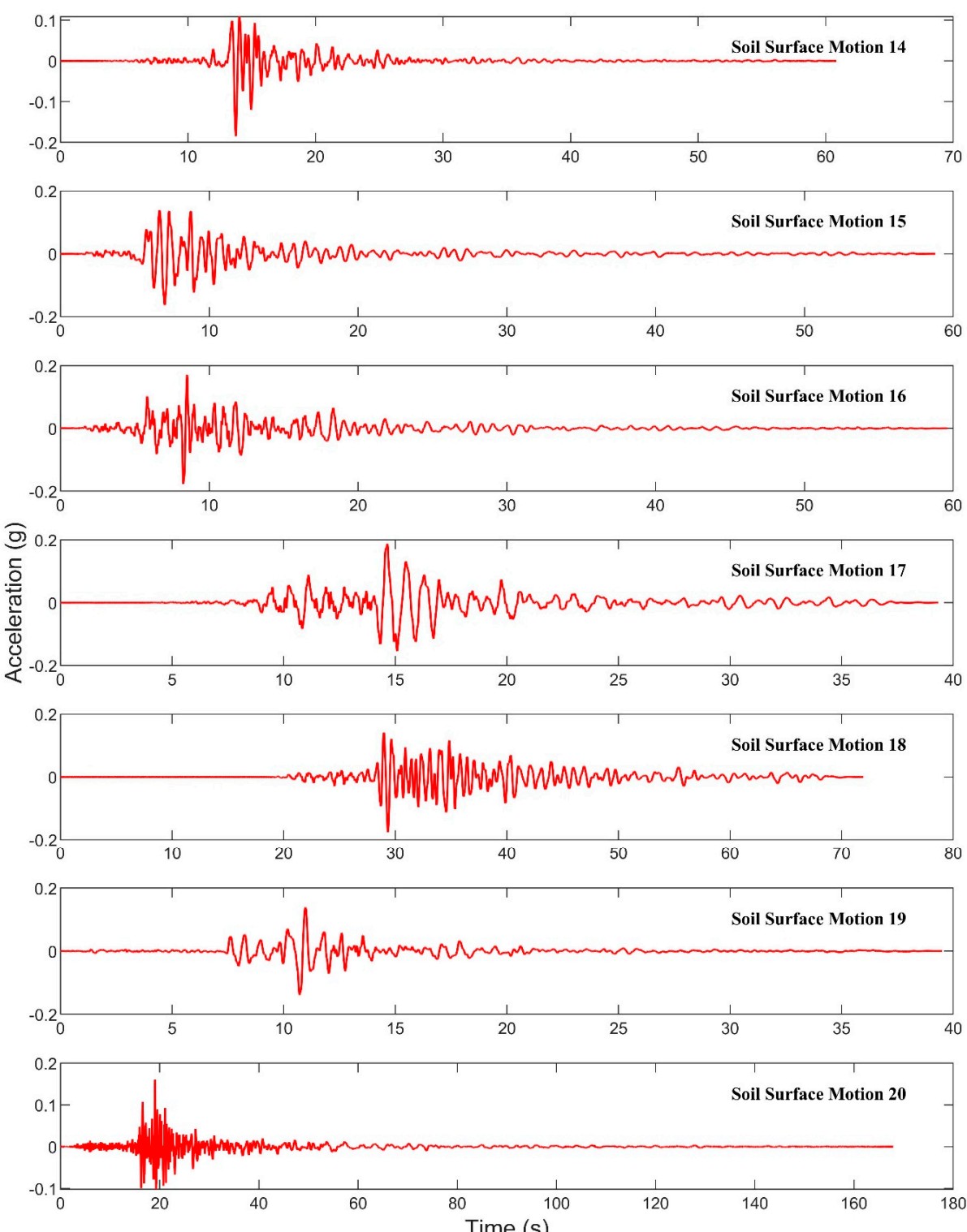

**Figure A2.** Soil surface accelerograms that are recommended for use in nonlinear time history analysis for structure no. 2 for the case study.

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
