# Peer review of "Generation of Site-Specific Accelerograms and Response Spectra Involving Sampling Information from Borehole Records"

_2673-4109, doi:10.3390/civileng4030046_

Round 1
Reviewer 1 Report
Please refer to my comments in the attached document.

Reviewer 2 Report
The paper is well-written. The topic described will be very much useful for both the academics as well as for the practicing engineers. I have a couple of clarifications:
1. Table 2: The references of the relationships should be provided.
2. Along with Hardin & Drnevich model, is the degradation of shear modulus of soil with progressive number of loading cycles considered?
Round 2
Reviewer 1 Report
Dear Authors,
Thank you for the revisions and response to my comments. My questions and comments are answered.